# Revisiting Non-Acyclic GFlowNets in Discrete Environments

**Nikita Morozov** [* 1]  **Ian Maksimov** [* 1]  **Daniil Tiapkin** [2 3]  **Sergey Samsonov** [1]

## Abstract

Generative Flow Networks (GFlowNets) are a family of generative models that learn to sample objects from a given probability distribution, potentially known up to a normalizing constant. Instead of working in the object space, GFlowNets proceed by sampling trajectories in an appropriately constructed directed acyclic graph environment, greatly relying on the acyclicity of the graph. In our paper, we revisit the theory that relaxes the acyclicity assumption and present a simpler theoretical framework for non-acyclic GFlowNets in discrete environments. Moreover, we provide various novel theoretical insights related to training with fixed backward policies, the nature of flow functions, and connections between entropy-regularized RL and non-acyclic GFlowNets, which naturally generalize the respective concepts and theoretical results from the acyclic setting. In addition, we experimentally re-examine the concept of loss stability in non-acyclic GFlowNet training, as well as validate our own theoretical findings.

## 1. Introduction

Generative Flow Networks (GFlowNets, Bengio et al., 2021) are models that aim to sample discrete objects from distributions known proportionally up to a constant. They operate by constructing an object through a sequence of stochastic transitions defined by a forward policy. GFlowNets have been successfully applied in various areas, starting from molecule generation (Bengio et al., 2021; Shen et al., 2024; Koziarski et al., 2024; Cretu et al., 2025) and biological sequence design (Jain et al., 2022; Kim et al., 2024) to combinatorial optimization (Zhang et al., 2023a;b; Kim et al.,

2025) and fine-tuning of large language models and diffusion models (Hu et al., 2023; Venkatraman et al., 2024; Uehara et al., 2024; Zhang et al., 2025; Lee et al., 2025). The detailed theoretical foundations of GFlowNets in discrete environments were developed in (Bengio et al., 2023). While the majority of GFlowNet literature considers the discrete setting, it is possible to apply the methodology of continuous GFlowNets (Lahlou et al., 2023) to sampling problems on more general spaces.

The main idea behind the generation process in GFlowNets lies in sampling trajectories in the appropriately constructed directed acyclic graph environment instead of working directly in the object space. A standard intuition behind this process is a sequence of actions applied in order to construct a composite object from "blocks". One of the limitations of this setting is that it requires acyclicity. While this limitation can be naturally interpreted in, e.g., molecule generation setting, it can confine the practical design of GFlowNet environments, as well as restrict the applicability of GFlowNets in other scenarios. A motivational example for non-acyclic GFlowNets presented by (Brunswic et al., 2024) is related to modeling distributions over objects with intrinsic symmetries. Consider a class of environments where states are elements of some group, e.g. symmetric group or Rubik's Cube group. The transitions are given via a generating set of this group, thus corresponding to applying the group operation on the current state and some element of the generating set, which leads to the existence of cycles. While in some cases an acyclic environment can be designed to generate group elements, such environments of "algebraic" origin naturally contain cycles, thus falling under the area of our study. In addition, there is a growing body of work that connects GFlowNets and Reinforcement Learning (Tiapkin et al., 2024; Mohammadpour et al., 2024; Deleu et al., 2024; He et al., 2024). Most RL environments contain cycles, thus understanding how GFlowNets can be applied in cyclic environments and connecting them to RL in such cases can be a crucial step towards further bridging two research fields.

To the best of our knowledge, methodological aspects of working with non-acyclic environments in GFlowNets were previously considered only in the recent work of (Brunswic et al., 2024). The latter paper, similarly to (Lahlou et al., 2023), uses the machinery of measurable spaces and measure theory, which is harder to build new extensions and

---

[*]Equal contribution [1]HSE University, Moscow, Russia [2]CMAP – CNRS – École polytechnique – Institut Polytechnique de Paris, 91128, Palaiseau, France [3]Université Paris-Saclay, CNRS, LMO, 91405, Orsay, France. Correspondence to: Nikita Morozov <nvmorozov@hse.ru>.

*Proceedings of the 42nd International Conference on Machine Learning*, Vancouver, Canada. PMLR 267, 2025. Copyright 2025 by the author(s).

methodology upon. We believe that simplicity is a key merit of the theory behind discrete GFlowNets (Bengio et al., 2023) when compared to their general state counterparts. Thus, the main aim of our paper is to revisit the theory of non-acyclic GFlowNets within a discrete state space, simplifying the constructions of (Brunswic et al., 2024) and providing additional theoretical and methodological insights into training GFlowNets in this setting. The main contributions of the paper can be summarized as follows:

1. We present a simple and intuitive way to build a theory of non-acyclic GFlowNets in discrete environments from scratch. In addition to simplicity, our construction introduces and clarifies a number of key points regarding similarities and dissimilarities between acyclic and non-acyclic discrete GFlowNets that were not explored in (Brunswic et al., 2024), in particular regarding the nature of flows and importance of backward policies.

2. We show that when the backward policy is fixed, the loss stability introduced by (Brunswic et al., 2024) does not impact the result of the optimization procedure. The latter becomes important only when the backward policy is also being trained.

3. When backward policy is trained, we show that learning a non-acyclic GFlowNet with the smallest expected trajectory length is equivalent to learning a non-acyclic GFlowNet with the smallest total flow. In addition, we propose state flow regularization as a way to approach the arising optimization problem.

4. We empirically show that the scale in which flow error is computed in the loss plays a crucial role in its stability. However, we also show that using an unstable loss with the proposed state flow regularization can lead to better sampling quality.

5. Finally, we generalize the key theoretical result of (Tiapkin et al., 2024) on the equivalence between GFlowNets and entropy-regularized RL to the non-acyclic setting.

Source code: github.com/GreatDrake/non-acyclic-gfn.

## 2. Background

### 2.1. GFlowNets

This section presents necessary notations and theoretical background on GFlowNets. GFlowNets treat the problem of sampling from a probability distribution over discrete space $\mathcal{X}$ as a sequential decision-making process in a directed acyclic graph (DAG) $\mathcal{G} = (\mathcal{S}, \mathcal{E})$, where $\mathcal{S}$ is a finite state space and $\mathcal{E} \subseteq \mathcal{S} \times \mathcal{S}$ is a finite set of edges (or transitions). There is a special *initial state* $s_0$ with no incoming edges and a special *sink state* $s_f$ with no outgoing edges. The commonly used variant of notation does not include a sink state $s_f$, yet we prefer to use a variant with $s_f$, since it was also used in the previous work on non-acyclic GFlowNets (Brunswic et al., 2024) and leads to a

more intuitive construction. Let $\mathcal{T}$ be a set of all trajectories $\tau = (s_0 \to s_1 \to \ldots \to s_{n_\tau} \to s_f)$ from $s_0$ to $s_f$, where we use $n_\tau$ to denote the length of the trajectory $\tau$. We use a convention $s_{n_\tau+1} = s_f$. We say that $\tau$ *terminates* in $s$ if its last transition is $s \to s_f$. Such transitions are called *terminating*, and the states that have an outgoing edge into $s_f$ are called *terminal* states. The set of terminal states coincides with $\mathcal{X}$, and the probability distribution of interest $\mathcal{R}(x)/\mathcal{Z}$ is defined on it, where $\mathcal{R}(x) > 0$ is called *GFlowNet reward* and $\mathcal{Z} = \sum_{x \in \mathcal{X}} \mathcal{R}(x)$ is an unknown normalizing constant. In addition, for any state $s$, we denote $\text{in}(s)$ to be the set of states $s'$ such that there is an edge $s' \to s \in \mathcal{E}$ (parents), and $\text{out}(s)$ to be the set of states $s'$ such that there is an edge $s \to s' \in \mathcal{E}$ (children).

The main goal of GFlowNets is to find a distribution $\mathcal{P}$ over $\mathcal{T}$ such that for any $x \in \mathcal{X}$, probability that $\tau \sim \mathcal{P}$ terminates in $x$ coincides with $\mathcal{R}(x)/\mathcal{Z}$. This property is called the *reward matching condition*. GFlowNets operate with Markovian distributions over trajectories (see (Bengio et al., 2023) for a definition and discussion) using the following key components:

1. a *forward policy* $\mathcal{P}_{\text{F}}(s' \mid s)$, which is a distribution over children of each state;

2. a *backward policy* $\mathcal{P}_{\text{B}}(s \mid s')$, which is a distribution over parents of each state;

3. *state/edge flows* $\mathcal{F}(s)$, $\mathcal{F}(s \to s')$, which coincide with an unnormalized probability that a trajectory $\tau$ passes through state/edge.

$\mathcal{P}_{\text{F}}, \mathcal{P}_{\text{B}}$, and $\mathcal{F}$ are connected through the *trajectory balance conditions*:

$$\mathcal{P}(\tau) = \prod_{t=0}^{n_\tau} \mathcal{P}_{\text{F}}(s_{t+1} \mid s_t) = \prod_{t=0}^{n_\tau} \mathcal{P}_{\text{B}}(s_t \mid s_{t+1}), \quad (1)$$

*detailed balance conditions*:

$$\mathcal{F}(s \to s') = \mathcal{F}(s)\mathcal{P}_{\text{F}}(s' \mid s) = \mathcal{F}(s')\mathcal{P}_{\text{B}}(s \mid s'), \quad (2)$$

and *flow matching conditions*:

$$\mathcal{F}(s) = \sum_{s' \in \text{out}(s)} \mathcal{F}(s \to s') = \sum_{s'' \in \text{in}(s)} \mathcal{F}(s'' \to s). \quad (3)$$

All of these objects are completely and uniquely specified if one fixes either 1) edge flow $\mathcal{F}(s \to s')$, 2) initial flow $\mathcal{F}(s_0)$ and $\mathcal{P}_{\text{F}}$, 3) initial flow $\mathcal{F}(s_0)$ and $\mathcal{P}_{\text{B}}$. If flows satisfy $\mathcal{F}(s \to s_f) = \mathcal{R}(s)$, trajectory distribution defined by the corresponding forward policy will satisfy the reward matching condition (Bengio et al., 2023). In practice, a neural network is used to parameterize the forward policy (and, optionally, the backward policy and the flows). Then, it is

trained to minimize some loss function that would enforce the reward matching condition. For example, *Detailed Balance* loss (Bengio et al., 2023) is defined on all transitions $s \to s' \in \mathcal{E}$ as:

$$\mathcal{L}_{\mathrm{DB}}(s \to s') \triangleq \left( \log \frac{\mathcal{F}_\theta(s)\mathcal{P}_{\mathrm{F}}(s' \mid s, \theta)}{\mathcal{F}_\theta(s')\mathcal{P}_{\mathrm{B}}(s \mid s', \theta)} \right)^2 . \quad (4)$$

Reward matching is enforced by substituting $\mathcal{F}(x \to s_f) = \mathcal{F}_\theta(s_f)\mathcal{P}_{\mathrm{B}}(x \mid s_f, \theta) = \mathcal{R}(x)$ in the loss. Although the optimization task typically admits multiple solutions, fixing the backward policy results in a unique solution in terms of $\mathcal{F}$ and $\mathcal{P}_{\mathrm{F}}$ (Bengio et al., 2023).

## 2.2. GFlowNets in Non-Acyclic Environments

(Brunswic et al., 2024) state that fundamental results of GFlowNet theory also apply in the case when the environment graph $\mathcal{G}$ may contain cycles, and all definitions from the acyclic case remain valid and extend to the non-acyclic case. However, we will further show that this is not exactly true, e.g., *flows cannot be consistently defined as unnormalized visitation probabilities*.

More specifically, (Brunswic et al., 2024) argue that if (3) holds for an edge flow, as well as $\mathcal{F}(s \to s_f) = \mathcal{R}(s)$ for terminating transitions, the forward policy induced by the flow $\mathcal{P}_{\mathrm{F}}(s' \mid s) = \frac{\mathcal{F}(s \to s')}{\mathcal{F}(s)}$ satisfies the reward matching condition. Thus, standard GFlowNet losses, such as Flow Matching (FM, Bengio et al., 2021), Detailed Balance (DB, Bengio et al., 2023), and Trajectory Balance (TB, Malkin et al., 2022) can be applied in non-acyclic environments.

However, (Brunswic et al., 2024) point out that the main distinction between non-acyclic and acyclic GFlowNets is that in the non-acyclic setting, expected trajectory length $\mathbb{E}[n_\tau]$ (denoted as a sampling time in (Brunswic et al., 2024)) can be arbitrarily large because of the cycles, while in the acyclic setting it is always bounded. To tackle this issue, a concept of *loss stability* is introduced. A loss is called *stable* if adding a constant to the flow along a cycle can never decrease the loss, and otherwise, it is called *unstable* (Definition 3). It is shown that FM, DB, and TB are unstable (Theorem 3), which can lead to arbitrarily large sampling time when utilized for training. In contrast, a family of losses that are provably stable is presented (Theorem 4). Moreover, the authors show that there are stable variants of FM and DB losses, such as stable DB loss, which we denote as SDB:

$$\mathcal{L}_{\mathrm{SDB}}(s \to s') \triangleq \log(1 + \varepsilon\Delta^2(s, s', \theta))(1 + \eta\mathcal{F}_\theta(s)),$$
$$\Delta(s, s', \theta) \triangleq \mathcal{F}_\theta(s)\mathcal{P}_{\mathrm{F}}(s'|s, \theta) - \mathcal{F}_\theta(s')\mathcal{P}_{\mathrm{B}}(s|s', \theta), \quad (5)$$

where $\varepsilon$ and $\eta$ are hyperparameters. In addition, (Brunswic et al., 2024) show that the expected trajectory length is

bounded by the total normalized state flow

$$\mathbb{E}[n_\tau] \leq \frac{1}{\mathcal{F}(s_0)} \sum_{s \in \mathcal{S} \setminus \{s_0, s_f\}} \mathcal{F}(s), \quad (6)$$

and using a stable loss with a regularizer that controls the total flow, e.g., the norm of the edge flow matrix, can be used to learn an acyclic flow (Theorem 1).

## 3. Theory of GFlowNets in Discrete Non-Acyclic Environments

### 3.1. Environment

All definitions regarding the environment can be introduced similarly to the setting of acyclic GFlowNets (see Section 2.1) with one main difference: graph $\mathcal{G}$ can now contain cycles. In addition to finiteness, we make two technical assumptions on the structure of $\mathcal{G}$:

**Assumption 3.1.** 0) graph $\mathcal{G}$ is finite; 1) There is a special initial state $s_0$ with no incoming edges and a special sink state $s_f$ with no outgoing edges; 2) For any state $s \in \mathcal{S}$ there exists a path from $s_0$ to $s$ and a path from $s$ to $s_f$.

Next, we formally define trajectories:

**Definition 3.2.** A sequence $\tau = (s_0 \to s_1 \to \ldots \to s_{n_\tau} \to s_{n_\tau+1} = s_f)$ is called a trajectory of length $n_\tau \in \mathbb{N}$ if all transitions $s_t \to s_{t+1} \in \mathcal{E}, t \in \{0, \ldots, n_\tau\}$. Then $\mathcal{T}$ is a set of all finite-length trajectories that start in $s_0$ and finish in $s_f$.

In the above definition and further, we use a convention $s_{n_\tau+1} = s_f$. While $\mathcal{G}$ itself is always finite, the main difference with acyclic GFlowNets is that $\mathcal{T}$ can be infinite, and $\mathcal{T}$ can contain trajectories of arbitrary length.

### 3.2. Backward Policy and Trajectory Distribution

There are several equivalent ways to introduce probability distributions over trajectories in GFlowNets. One of the common approaches is to start by introducing trajectory flows (Bengio et al., 2023). The main theoretical advantage of the approach based on trajectory flows is that it allows for non-Markovian flows, see (Bengio et al., 2023). At the same time, Markovian flows are primarily considered in GFlowNets literature, and in our paper, we only consider this setting. Instead of starting from the definition of the flow, a more intuitive approach is to begin with the definitions of the *forward* and *backward* policies.

**Definition 3.3.** A forward policy $\mathcal{P}_{\mathrm{F}}(s' \mid s)$ consistent with $\mathcal{G}$ is a family of conditional probability distributions over $s' \in \mathrm{out}(s)$ defined for each $s \in \mathcal{S} \setminus \{s_f\}$, Similarly, a backward policy $\mathcal{P}_{\mathrm{B}}(s \mid s')$ consistent with $\mathcal{G}$ is a family of conditional probability distribution over $s \in \mathrm{in}(s')$, defined for each $s' \in \mathcal{S} \setminus \{s_0\}$.

In the subsequent parts of the paper, we always assume that the considered $\mathcal{P}_{\mathrm{F}}$ or $\mathcal{P}_{\mathrm{B}}$ are consistent with $\mathcal{G}$ and do not specify this fact explicitly. Definition 3.3 is consistent with the definitions of forward and backward policies in acyclic GFlowNets (Bengio et al., 2023, Definition 4). Note that the structure of $\mathcal{G}$ is symmetric with respect to an interchange between initial state $s_0$ and sink state $s_f$ and reversion of all edges in $\mathcal{G}$. Thus, starting with either $\mathcal{P}_{\mathrm{F}}$ or $\mathcal{P}_{\mathrm{B}}$ is equivalent. We prefer to start from a backward policy $\mathcal{P}_{\mathrm{B}}$ in our subsequent derivations. Using $\mathcal{P}_{\mathrm{B}}$, we define a probability distribution $\mathcal{P}$ over $\tau = (s_0 \to s_1 \to \ldots \to s_{n_\tau} \to s_f) \in \mathcal{T}$ according to

$$\mathcal{P}(\tau) \triangleq \prod_{t=0}^{n_\tau} \mathcal{P}_{\mathrm{B}}(s_t \mid s_{t+1}). \tag{7}$$

In such a case, we say that the trajectory distribution $\mathcal{P}(\tau)$ is *induced by* $\mathcal{P}_{\mathrm{B}}$. In the following lemma, we show that $\mathcal{P}(\tau)$ is a correctly defined probability distribution over $\mathcal{T}$.

**Lemma 3.4.** *Let $\mathcal{P}_{\mathrm{B}}(s \mid s')$ be a backward policy, such that $\mathcal{P}_{\mathrm{B}}(s \mid s') > 0$ for any edge $s \to s' \in \mathcal{E}$. Then*

- *$\mathcal{P}(\tau)$ defined in (7) is a probability distribution over $\mathcal{T}$, that is, $\sum_{\tau \in \mathcal{T}} \mathcal{P}(\tau) = 1$.*
- *Moreover, its expected trajectory length is always finite $\mathbb{E}_{\tau \sim \mathcal{P}}[n_\tau] = \sum_{\tau \in \mathcal{T}} n_\tau \mathcal{P}(\tau) < +\infty$.*

In fact, the condition $\mathcal{P}_{\mathrm{B}}(s \mid s') > 0$ together with Assumption 3.1 allows us to ensure that the sequence $s_i$ is a finite state-space absorbing Markov chain. Given this assumption, the proof of Lemma 3.4 almost coincides with the proof of the fact that the states of such a Markov chain are transient, see, e.g., (Douc et al., 2018). For completeness, we provide the proof in Appendix A.1.

### 3.3. State and Edge Flows

Given a probability distribution $\mathcal{P}(\tau)$ induced by $\mathcal{P}_{\mathrm{B}}$, our next aim is to define state and edge flows. Before proceeding with a valid construction, we provide some intuition about our definitions. Let us first show that, contrary to the acyclic GFlowNets, we cannot define edge flows as *visitation probabilities* $\mathcal{P}(\{\tau \in \mathcal{T} \mid s \to s' \in \tau\})$. In particular, we show that such a definition does not satisfy the flow matching conditions (3). Indeed, consider an example from (Brunswic et al., 2024):

$$s_0 \xrightarrow{1} a \xrightarrow{0.5} b \overset{0.5}{\underset{1}{\rightleftarrows}} c \xrightarrow{1} s_f$$

The number above each edge $s \to s'$ is $\mathcal{P}_{\mathrm{B}}(s \mid s')$. Consider the distribution $\mathcal{P}(\tau)$ defined by (7). Let us plot the visitation probability for each edge:

$$s_0 \xrightarrow{1} a \xrightarrow{1} b \overset{0.5}{\underset{1}{\rightleftarrows}} c \xrightarrow{1} s_f$$

One can see that the flow matching condition (3) does not hold for states $b$ and $c$ since $1 \neq 1 + 0.5$. Instead, let us calculate the *expected number of visits* for each edge $s \to s'$

$$\mathbb{E}_{\tau \sim \mathcal{P}}\left[\sum_{t=0}^{n_\tau} \mathbb{I}\{s_t = s, s_{t+1} = s'\}\right].$$

We visualize the corresponding numbers on the plot below:

$$s_0 \xrightarrow{1} a \xrightarrow{1} b \overset{1}{\underset{2}{\rightleftarrows}} c \xrightarrow{1} s_f$$

It is easy to check that the flow matching conditions (3) are now satisfied. Next, we formally define:

**Definition 3.5.** Let $\mathcal{P}_{\mathrm{B}}(s \mid s')$ be a backward policy, such that $\mathcal{P}_{\mathrm{B}}(s \mid s') > 0$ for any edge $s \to s' \in \mathcal{E}$. Then, given a *final flow* $\mathcal{F}(s_f) > 0$, we define state and edge flows as

$$\mathcal{F}(s \to s') \triangleq \mathcal{F}(s_f) \cdot \mathbb{E}_{\tau \sim \mathcal{P}}\left[\sum_{t=0}^{n_\tau} \mathbb{I}\{s_t = s, s_{t+1} = s'\}\right],$$

$$\mathcal{F}(s) \triangleq \mathcal{F}(s_f) \cdot \mathbb{E}_{\tau \sim \mathcal{P}(\tau)}\left[\sum_{t=0}^{n_\tau+1} \mathbb{I}\{s_t = s\}\right]. \tag{8}$$

We say that the flows defined above are induced by the backward policy $\mathcal{P}_{\mathrm{B}}$ and final flow $\mathcal{F}(s_f)$.

It is important to note that if $\mathcal{G}$ does not contain cycles, the expected number of visits in (8) coincides with visitation probability, thus Definition 3.5 agrees with the usual understanding of flows in the acyclic GFlowNet literature. Next, we show that state and edge flows defined in (3.5) satisfy the detailed balance and flow matching conditions (2) – (3).

**Proposition 3.6.** *Flows $\mathcal{F}$ defined in Definition 3.5 satisfy:*

1. *$\mathcal{F}(s) \overset{(a)}{=} \sum_{s' \in \mathrm{out}(s)} \mathcal{F}(s \to s') \overset{(b)}{=} \sum_{s'' \, \mathrm{in}(s)} \mathcal{F}(s'' \to s)$, for each $s \in \mathcal{S} \setminus \{s_0, s_f\}$. Moreover, identity (a) holds for $s_0$, and (b) holds for $s_f$.*

2. *$\mathcal{F}(s \to s') = \mathcal{F}(s')\mathcal{P}_{\mathrm{B}}(s \mid s')$ for any $s \to s' \in \mathcal{E}$.*

3. *$\mathcal{F}(s_0) = \mathcal{F}(s_f)$.*

We provide the proof in Appendix A.2. In the next proposition, we show that there is a one-to-one correspondence between a pair $(\mathcal{P}_{\mathrm{B}}, \mathcal{F}(s_f))$ and edge flows $\mathcal{F}$. Its proof is provided in Appendix A.3.

**Proposition 3.7.** *Let $\mathcal{F} : \mathcal{E} \to \mathbb{R}_{>0}$ be a function that satisfies the flow matching conditions (3). Define the corresponding backward policy by the relation*

$$\mathcal{P}_{\mathrm{B}}(s \mid s') = \mathcal{F}(s \to s') \,\Big/ \sum_{s'' \in \mathrm{in}(s')} \mathcal{F}(s'' \to s').$$

*Then $\mathcal{F}$ are edge flows induced by $\mathcal{P}_{\mathrm{B}}$ and $\mathcal{F}(s_f) = \sum_{s'' \in \mathrm{in}(s_f)} \mathcal{F}(s'' \to s_f)$.*

## 3.4. Forward Policy and Detailed Balance

It is well-known in acyclic GFlowNets theory (Bengio et al., 2023) that there exists a unique forward policy $\mathcal{P}_F$ for any backward policy $\mathcal{P}_B$ that induces the same probability distribution over $\mathcal{T}$. The main implication of this fact is that by fixing rewards $\mathcal{R}(x), x \in \mathcal{X}$ and a backward policy $\mathcal{P}_B(s \mid s')$ for each state $s' \in \mathcal{S} \setminus \{s_0, s_f\}$, one automatically fixes a trajectory distribution $\mathcal{P}(\tau)$ that satisfies the reward matching condition (Malkin et al., 2022). However, sampling from a such distribution is intractable since it requires starting from a terminal state sampled from the reward distribution. Thus, during GFlowNet training, one tries to find a forward policy, which always allows tractable sampling of trajectories by construction, that will match this trajectory distribution $\mathcal{P}(\tau)$. One can note that this bears similarities with hierarchical variational inference (Malkin et al., 2023). In the following proposition, we show that this result also holds for non-acyclic GFlowNets.

**Proposition 3.8.** *Given any backward policy $\mathcal{P}_B(s \mid s') > 0$, there exists a unique forward policy $\mathcal{P}_F(s' \mid s)$ such that*

$$\mathcal{P}(\tau) = \prod_{t=0}^{n_\tau} \mathcal{P}_B(s_t \mid s_{t+1}) = \prod_{t=0}^{n_\tau} \mathcal{P}_F(s_{t+1} \mid s_t), \ \forall \tau \in \mathcal{T}.$$

*Moreover, it satisfies the detailed balance conditions*

$$\mathcal{F}(s)\mathcal{P}_F(s' \mid s) = \mathcal{F}(s')\mathcal{P}_B(s \mid s'), \ \forall s \to s' \in \mathcal{E}$$

*with the state flow $\mathcal{F}$ defined in (8).*

The proof is provided in Appendix A.4. Conversely, the following proposition shows that if a triplet $\mathcal{F}, \mathcal{P}_F, \mathcal{P}_B$ satisfies the detailed balance conditions (2), it will be consistent with all previous definitions and propositions.

**Proposition 3.9.** *Let $\mathcal{F}\colon \mathcal{S} \to \mathbb{R}_{>0}$, and let $\mathcal{P}_F(s'|s) > 0$, $\mathcal{P}_B(s|s') > 0$ be forward and backward policies, such that the detailed balance conditions (2) are satisfied. Then $\mathcal{P}_F$ and $\mathcal{P}_B$ induce the same trajectory distribution:*

$$\mathcal{P}(\tau) = \prod_{t=0}^{n_\tau} \mathcal{P}_B(s_t \mid s_{t+1}) = \prod_{t=0}^{n_\tau} \mathcal{P}_F(s_{t+1} \mid s_t), \ \forall \tau \in \mathcal{T}.$$

*Moreover, then $\mathcal{F}$ are state flows induced by $\mathcal{P}_B$ and $\mathcal{F}(s_f)$.*

For proof, we refer to Appendix A.5. The above propositions directly generalize their counterparts from the non-acyclic setting (Bengio et al., 2023). Note that, due to the symmetries between $s_0$ and $s_f$ in $\mathcal{G}$ up to changing direction of edges, we could start from the forward policy and trajectory distribution induced by it in (7), and then prove the uniqueness of the corresponding backward policy $\mathcal{P}_B$.

## 3.5. Training Non-Acyclic GFlowNets

Now, we proceed with the main learning problem in GFlowNets: finding a forward policy that matches the reward distribution over terminal states $\mathcal{R}(x)/\mathcal{Z}, x \in \mathcal{X}$. The

following proposition shows how the reward matching condition can be formulated in terms of flows.

**Proposition 3.10.** *Let $\mathcal{P}_B(s \mid s') > 0$ be a backward policy, $\mathcal{F}(s_f) > 0$ a final flow, and $\mathcal{R}(x) > 0$ GFlowNet rewards. If edge flows $\mathcal{F}(s \to s')$ induced by $\mathcal{P}_B$ and $\mathcal{F}(s_f)$ satisfy:*

$$\mathcal{F}(x \to s_f) = \mathcal{R}(x) \ \forall x \to s_f \in \mathcal{E}, \qquad (9)$$

*the trajectory distribution $\mathcal{P}$ induced by $\mathcal{P}_B$ (7) satisfies the reward matching condition, i.e. $\mathbb{P}_{\tau \sim \mathcal{P}}[s_{n_\tau} = x] = \mathcal{R}(x)/\mathcal{Z}$. Then, the same trajectory distribution is induced by the unique corresponding forward policy $\mathcal{P}_F$, thus also satisfying the reward matching condition.*

*Proof.* Notice that an edge leading into $s_f$ can be visited only once; thus, $\mathcal{F}(x \to s_f)$ coincides with a probability $\mathbb{P}_{\tau \sim \mathcal{P}}[s_{n_\tau} = x]$ that a trajectory terminates in $x$ times the final flow $\mathcal{F}(s_f)$. In addition, we have $\mathcal{F}(s_f) = \sum_{x \in \text{in}(s_f)} \mathcal{F}(x \to s_f) = \sum_{x \in \mathcal{X}} \mathcal{R}(x) = \mathcal{Z}$, thus $\mathbb{P}_{\tau \sim \mathcal{P}}[s_{n_\tau} = x] = \mathcal{F}(x \to s_f)/\mathcal{F}(s_f) = \mathcal{R}(x)/\mathcal{Z}$. $\square$

Proposition 3.10 also implies $\mathcal{F}(s_0) = \mathcal{F}(s_f) = \mathcal{Z}$ and $\mathcal{P}_B(x \mid s_f) = \mathcal{R}(x)/\mathcal{Z}$ by Proposition 3.6.

An important fact from the literature on acyclic GFlowNets (Malkin et al., 2022; Bengio et al., 2023) that was overlooked in the work of (Brunswic et al., 2024), but holds in the non-acyclic case as well, is that it is generally easy to manually pick a backward policy such that the induced trajectory distribution will satisfy the reward matching condition. A simple and natural choice is to take $\mathcal{P}_B(x \mid s_f) = \mathcal{R}(x)/\mathcal{Z}$ for $s_f$ and fix $\mathcal{P}_B(s \mid s') = 1/|\text{in}(s')|$ for all other states. It is worth mentioning that $\mathcal{Z}$ is generally unknown, but this issue is circumvented in GFlowNets by learning unnormalized flows or making $\mathcal{Z}$ itself a learnable parameter depending on the chosen loss function (Malkin et al., 2022; Bengio et al., 2023; Madan et al., 2023). Moreover, Proposition 3.8 shows the uniqueness of the corresponding $\mathcal{P}_F$. Thus, we state the main practical corollary of this result:

**Corollary 3.11.** *When a backward policy $\mathcal{P}_B > 0$ is fixed, any loss from the acyclic GFlowNet literature (Bengio et al., 2021; Malkin et al., 2022; Bengio et al., 2023; Madan et al., 2023) can be used to learn the corresponding forward policy $\mathcal{P}_F$ in the non-acyclic case as well. Lemma 3.4 and Proposition 3.8 imply that there is always a unique solution with a finite expected trajectory length, thus the stability of the loss (Brunswic et al., 2024) does not play a factor.*

The main disadvantage of learning with a fixed backward policy in non-acyclic GFlowNets that does not arise in acyclic GFlowNets is the fact that the expected trajectory length $\mathbb{E}_{\tau \sim \mathcal{P}}[n_\tau]$ of a manually chosen $\mathcal{P}_B$ can be large. A natural way to circumvent this issue is to consider a

learnable backward policy, which is also a widely employed choice in acyclic GFlowNet literature (Malkin et al., 2022; Jang et al., 2024a; Gritsaev et al., 2025). However, (Brunswic et al., 2024) made an important discovery by pointing out that standard losses from acyclic GFlowNet literature are not stable (Theorem 3), meaning that the expected trajectory length can grow uncontrollably during training. The concept of stability was introduced with respect to learnable edge flows (Definition 3), which implies that the corresponding backward policy also changes during training. Using a stable loss, e.g. (5), was proposed as a way to approach this issue. At the same time, we argue that efficient training of a non-acyclic GFlowNet with controlled expected trajectory length in case of a learnable $\mathcal{P}_\mathrm{B}$ is possible without utilizing stable losses. The next proposition is a simple corollary of Definition 3.5:

**Proposition 3.12.** *Given a trajectory distribution $\mathcal{P}$, its expected trajectory length is equal to the normalized total flow:*

$$\mathbb{E}_{\tau \sim \mathcal{P}}[n_\tau] = \frac{1}{\mathcal{F}(s_f)} \sum_{s \in \mathcal{S} \setminus \{s_0, s_f\}} \mathcal{F}(s). \qquad (10)$$

The proof is presented in Appendix A.6. This result is a refinement of Theorem 2 of (Brunswic et al., 2024), which states only '$\leq$' inequality in (10). Thus, we believe one of our key contributions to be pointing out the following fact:

> Learning a non-acyclic GFlowNet with the smallest expected trajectory length is *equivalent* to learning a non-acyclic GFlowNet with the smallest total flow.

We also believe that exploiting this equivalence is a crucial direction for future research on non-acyclic GFlowNets. We further explore a particular solution, which suggests the use of a state flow as a regularizer in the existing GFlowNet losses. We consider an example of the detailed balance loss DB (4). In this case, Proposition 3.9 implies that learning a non-acyclic GFlowNet with the smallest expected trajectory length can be formulated as the following constrained optimization problem:

$$\min_{\mathcal{F}, \mathcal{P}_\mathrm{F}, \mathcal{P}_\mathrm{B}} \sum_{s \in \mathcal{S} \setminus \{s_0, s_f\}} \mathcal{F}(s) \qquad (11)$$

$$\text{subject to } \left( \log \frac{\mathcal{F}(s) \mathcal{P}_\mathrm{F}(s' \mid s)}{\mathcal{F}(s') \mathcal{P}_\mathrm{B}(s \mid s')} \right)^2 = 0, \quad \forall s \to s' \in \mathcal{E},$$

$$\mathcal{F}(s_f) \mathcal{P}_\mathrm{B}(x \mid s_f) = \mathcal{R}(x), \qquad \forall x \to s_f \in \mathcal{E}.$$

As an approximate way to solve (11), one can use DB (4) with *state flow regularization*:

$$\left( \log \frac{\mathcal{F}_\theta(s) \mathcal{P}_\mathrm{F}(s' \mid s, \theta)}{\mathcal{F}_\theta(s') \mathcal{P}_\mathrm{B}(s \mid s', \theta)} \right)^2 + \lambda \mathcal{F}_\theta(s), \qquad (12)$$

where $\lambda > 0$ is a hyperparameter that controls a trade-off between an expected trajectory length and an accuracy of matching the reward distribution. As in (4), reward matching is enforced by substituting $\mathcal{F}_\theta(s_f) \mathcal{P}_\mathrm{B}(x \mid s_f, \theta) = \mathcal{R}(x)$.

Note that the DB loss is defined on individual transitions, and during training, it is optimized across transitions collected by a training policy. A standard choice is to optimize it over transitions from trajectories sampled using $\mathcal{P}_\mathrm{F}$, yet the training policy can be chosen differently, or, in RL terms, training can be done in an on-policy or off-policy fashion, see (Tiapkin et al., 2024). Note that different states $s$ might appear with different frequencies in the loss depending on a training policy, which can lead to a bias in the optimization problem (11). We discuss this phenomenon in more detail, as well as potential ways to mitigate it, in Appendix B.1.

Finally, it is important to mention that flow-based regularizers in the non-acyclic case were already proposed in Theorem 1 of (Brunswic et al., 2024), but only for the stable loss setup. Moreover, they were introduced in order to find an acyclic flow. Our paper further explores and sheds new light on this phenomenon, showing that training can be carried out even when an unstable loss is utilized with regularization. Moreover, when the total flow is minimized, one can ensure the smallest possible expected trajectory length. It is also worth pointing out that the idea of introducing a constrained optimization problem to accommodate for cycles in GFlowNets was mentioned in (Deleu, 2025).

### 3.6. Connections with Entropy-Regularized RL

A recent line of works (Tiapkin et al., 2024; Deleu et al., 2024) studied connections between GFlowNets and RL, showing that the GFlowNet learning problem is equivalent to an entropy-regularized RL (Neu et al., 2017; Geist et al., 2019) problem in an appropriately constructed deterministic MDP, given that the backward policy is fixed. We show that the same result holds for non-acyclic GFlowNets as well.

Let $\mathcal{G}$ be a graph of a non-acyclic GFlowNet, $\mathcal{R}$ a GFlowNet reward, and $\mathcal{P}_\mathrm{B} > 0$ a fixed backward policy that satisfies the reward matching condition. Let $\mathcal{F}$ be the flow induced by $\mathcal{P}_\mathrm{B}$ with $\mathcal{F}(s_f) = \mathcal{Z}$, and $\mathcal{P}_\mathrm{F}$ be a unique forward policy corresponding to $\mathcal{P}_\mathrm{B}$ (see Proposition 3.8). Define a deterministic MDP $\mathcal{M}_\mathcal{G}$ induced by a graph $\mathcal{G}$, where the state space $\mathcal{S}$ coincides with vertices of $\mathcal{G}$, the action space $\mathcal{A}_s$ for each state $s$ corresponds to $\mathrm{out}(s)$, and the transition kernel is defined as transition in the graph $\mathrm{P}(s' \mid s, a) = \mathbb{I}\{a = s'\}$, $a \in \mathrm{out}(s)$. We use no discounting ($\gamma = 1.0$) and set RL rewards for terminating transitions $r(x, s_f) = \log \mathcal{R}(x)$, and for all other transitions $r(s, s') = \log \mathcal{P}_\mathrm{B}(s \mid s')$. Then, the following statement holds.

**Theorem 3.13** (Generalization of Theorem 1 (Tiapkin et al., 2024)). *The optimal policy $\pi^\star_{\lambda=1}(s' \mid s)$ for the entropy-*

*regularized MDP $\mathcal{M}_{\mathcal{G}}$ with coefficient $\lambda = 1$ is equal to $\mathcal{P}_{\mathrm{F}}(s' \mid s)$ for all $s \in \mathcal{S} \setminus \{s_f\}, s' \in \mathcal{A}_s$. Moreover, regularized optimal value $V_{\lambda=1}^{\star}(s)$ and Q-value $Q_{\lambda=1}^{\star}(s, s')$ coincide with $\log \mathcal{F}(s)$ and $\log \mathcal{F}(s \to s')$ respectively for all $s \to s' \in \mathcal{E}$.*

The proof and all missing definitions are provided in Appendix A.7. Note that the proof of (Tiapkin et al., 2024) cannot be directly transferred to the non-acyclic setting since it is based on induction over the topological ordering of vertices of $\mathcal{G}$, which exists only for acyclic graphs.

# 4. Experiments

In addition to verifying our theoretical findings, one of the goals of our experimental evaluation is to examine the *scaling hypothesis* that we put out:

> **Scaling hypothesis.** When $\mathcal{P}_{\mathrm{B}}$ is trainable, the main factor that plays a crucial role in loss stability in practice, i.e., the controlled mean trajectory length of the trained non-acyclic GFlowNet, is the scale in which the error between flows is computed. Indeed, the standard DB loss (4) operates in log-flow scale $\Delta \log \mathcal{F}$, while standard SDB (5) operates in flow scale $\Delta \mathcal{F}$. We hypothesize that using log-flow scale losses without regularization can lead to arbitrarily large trajectory length, while flow scale losses are biased towards solutions with smaller flows and thus do not suffer from this issue.

In this section, we use DB or SDB to specify the utilized loss, $\Delta \log \mathcal{F}$ or $\Delta \mathcal{F}$ to specify the flow scale used to compute the error, and use $\lambda = C$ to specify the strength of the proposed state flow regularization (see Section 3.5). For example, $(\mathrm{DB}, \Delta \log \mathcal{F})$ in the legend corresponds to (4), $(\mathrm{SDB}, \Delta \mathcal{F})$ corresponds to (5) and $(\mathrm{DB}, \Delta \log \mathcal{F}, \lambda = C)$ corresponds to (12). Detailed discussion on loss scaling and stability is provided in Appendix B.2.

## 4.1. Experimental Setting

We consider two discrete environments for experimental evaluation: 1) a non-acyclic version of the hypergrid environment (Bengio et al., 2021) that was introduced in (Brunswic et al., 2024); 2) non-acyclic permutation generation environment from (Brunswic et al., 2024) with a harder variant of the reward function. Experimental details are presented in Appendix C.

Mean sample reward was used as a metric in (Brunswic et al., 2024), with higher values considered better. However, *we point out that this does not always represent sampling accuracy from the reward distribution $\mathcal{R}/\mathcal{Z}$. Indeed, the model that learned to sample from the highest-reward mode*

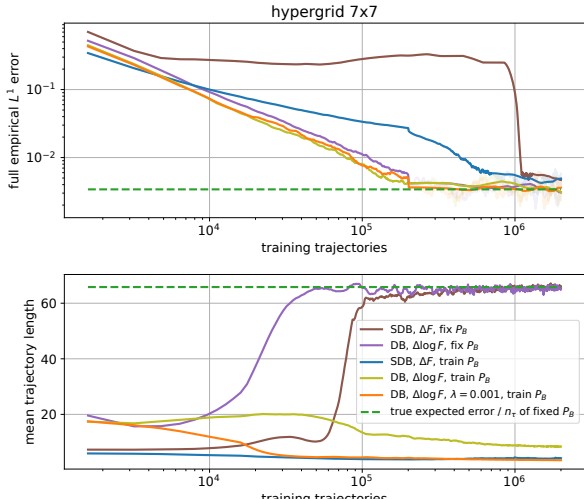

*Figure 1.* Comparison of non-acyclic GFlowNet training losses on a small hypergrid environment. We use DB or SDB to specify the utilized loss, $\Delta \log \mathcal{F}$ or $\Delta \mathcal{F}$ to specify the flow scale used to compute the error in the loss, and use $\lambda = C$ to specify the usage of the proposed state flow regularization. *Top:* evolution of $L^1$ distance between an empirical distribution of samples and target distribution. *Bottom:* evolution of mean length of sampled trajectories.

still achieves a high average reward despite resulting in mode collapse. For instance, recent works argue that measuring the deviation of mean sample reward from the true expected reward $\sum_{x \in \mathcal{X}} \mathcal{R}(x) \frac{\mathcal{R}(x)}{\mathcal{Z}}$ results in a better metric, see, e.g., (Shen et al., 2023) for detailed motivation. In addition, we employ other metrics to track sampling accuracy depending on the environment, which we discuss in detail below.

In both environments, we consider two settings: training with a fixed backward policy $\mathcal{P}_{\mathrm{B}}$ that is almost uniform and using a trainable $\mathcal{P}_{\mathrm{B}}$. In the second case, the initial log flow $\log \mathcal{F}_\theta(s_0) = \log \mathcal{Z}_\theta$ is also being learned. Thus, we can examine its convergence to the logarithm of the true normalizing constant $\log \mathcal{Z}$. See Appendix B.3 for details.

## 4.2. Hypergrids

We start with non-acyclic hypergrid environments (Brunswic et al., 2024). These environments are small enough that the normalizing constant $\mathcal{Z}$ can be computed exactly, and the trained sampler can be efficiently evaluated against the exact reward distribution. States are points with integer coordinates $s \in \{0, \ldots, H-1\}^D$ inside a $D$-dimensional hypercube with side length $H$, plus two auxiliary states $s_0$ and $s_f$. Possible transitions correspond to increasing or decreasing any coordinate by 1 without exiting the grid. Moreover, each state has a terminating

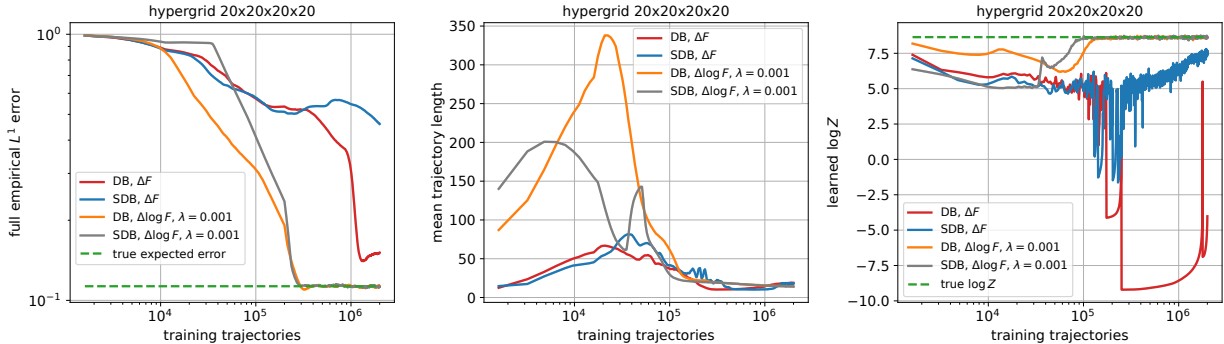

*Figure 2.* Comparison of non-acyclic GFlowNet training losses on a larger hypergrid environment. We use `DB` or `SDB` to specify the utilized loss, $\Delta \log \mathcal{F}$ or $\Delta \mathcal{F}$ to specify the flow scale used to compute the error in the loss, and use $\lambda = C$ to specify the usage of the proposed state flow regularization. *Left:* evolution of $L^1$ distance between an empirical distribution of samples and target distribution. *Middle:* evolution of mean length of sampled trajectories. *Right:* evolution of the trained initial log flow $\log \mathcal{Z}_\theta$.

*Table 1.* Comparison on the permutation environment. $C(k)\ L^1$ is the $L^1$ distance between the true and empirical distribution of fixed point probabilities $C(k)$, $\Delta \mathcal{R}$ is the relative error of mean reward proposed in (Shen et al., 2023), $\Delta \log \mathcal{Z}$ is $|\log \mathcal{Z}_\theta - \log \mathcal{Z}|$. Mean and std values are computed over 3 random seeds. $\boxed{\text{Blue}}$ indicates the best metric, $\boxed{\text{red}}$ indicates the smallest expected trajectory length.

| Loss | $n = 8$ | | | | $n = 20$ | | | |
|---|---|---|---|---|---|---|---|---|
| | $C(k)\ L^1 \downarrow$ | $\Delta \mathcal{R} \downarrow$ | $\Delta \log \mathcal{Z} \downarrow$ | $\mathbb{E}[n_\tau]$ | $C(k)\ L^1 \downarrow$ | $\Delta \mathcal{R} \downarrow$ | $\Delta \log \mathcal{Z} \downarrow$ | $\mathbb{E}[n_\tau]$ |
| DB, $\Delta \mathcal{F}$ | $0.215_{\pm 0.198}$ | $0.214_{\pm 0.086}$ | $0.814_{\pm 0.826}$ | $2.43_{\pm 0.28}$ | $0.453_{\pm 0.002}$ | $0.343_{\pm 0.000}$ | $42.98_{\pm 0.000}$ | $2.00_{\pm 0.00}$ |
| SDB, $\Delta \mathcal{F}$ | $0.031_{\pm 0.012}$ | $0.046_{\pm 0.023}$ | $0.074_{\pm 0.025}$ | $3.32_{\pm 0.15}$ | $0.452_{\pm 0.001}$ | $0.343_{\pm 0.000}$ | $42.98_{\pm 0.000}$ | $2.01_{\pm 0.00}$ |
| DB, $\Delta \log \mathcal{F}, \lambda = 10^{-3}$ | $0.036_{\pm 0.015}$ | $0.056_{\pm 0.024}$ | $0.018_{\pm 0.010}$ | $2.80_{\pm 0.04}$ | $0.041_{\pm 0.002}$ | $0.064_{\pm 0.000}$ | $0.023_{\pm 0.005}$ | $3.23_{\pm 0.00}$ |
| SDB, $\Delta \log \mathcal{F}, \lambda = 10^{-3}$ | $0.037_{\pm 0.013}$ | $0.056_{\pm 0.019}$ | $0.020_{\pm 0.015}$ | $2.79_{\pm 0.04}$ | $0.041_{\pm 0.002}$ | $0.064_{\pm 0.000}$ | $0.026_{\pm 0.003}$ | $3.22_{\pm 0.00}$ |
| DB, $\Delta \log \mathcal{F}, \lambda = 10^{-5}$ | $0.005_{\pm 0.001}$ | $0.001_{\pm 0.000}$ | $0.005_{\pm 0.004}$ | $4.31_{\pm 0.05}$ | $0.017_{\pm 0.002}$ | $0.035_{\pm 0.002}$ | $0.003_{\pm 0.003}$ | $7.55_{\pm 0.50}$ |
| SDB, $\Delta \log \mathcal{F}, \lambda = 10^{-5}$ | $0.005_{\pm 0.001}$ | $0.002_{\pm 0.000}$ | $0.006_{\pm 0.006}$ | $4.36_{\pm 0.09}$ | $0.014_{\pm 0.001}$ | $0.025_{\pm 0.001}$ | $0.005_{\pm 0.005}$ | $7.31_{\pm 0.07}$ |

transition $s \to s_f$, thus $\mathcal{X} = \mathcal{S} \setminus \{s_0, s_f\}$. The reward function has modes near the grid corners, separated by wide troughs with very small rewards. To measure sampling accuracy, total variation distance is computed between the true reward distribution $\mathcal{R}(x)/\mathcal{Z}$ and an empirical distribution of the last $2 \cdot 10^5$ samples seen in training, which coincides with $\frac{1}{2}$ of the $L^1$ distance on discrete domains.

We begin our analysis with a $7 \times 7$ grid to study the effects of learning under a fixed backward policy compared to a trainable backward policy. Since the environment is small, it is possible to find the flows induced by the fixed $\mathcal{P}_B$ exactly, thus also its expected trajectory length, see Appendix B.4. Figure 1 presents the results. First, we note that both $(\text{DB}, \Delta \log \mathcal{F})$ and $(\text{SDB}, \Delta \mathcal{F})$ with fixed $\mathcal{P}_B$ converge to the true expected trajectory length induced by the fixed backward policy, which is in line with Corollary 3.11. However, for all losses, using trainable $\mathcal{P}_B$ allows us to find a solution with a smaller trajectory length. In addition, we observe that using a loss in $\Delta \mathcal{F}$ scale results in slower convergence and a slight bias in the learned forward policy than in the case of $\Delta \log \mathcal{F}$ scale, for both fixed and learned

$\mathcal{P}_B$. Finally, an interesting note is that using an unstable `DB` loss in $\Delta \log \mathcal{F}$ scale without state flow regularization *can* still result in a small expected trajectory length, as we see in this experiment. However, we further show that this is not the case for a larger environment.

Next, we consider a larger $20 \times 20 \times 20 \times 20$ hypergrid. An expected trajectory length induced by the chosen fixed backward policy is several orders of magnitude larger than for a smaller grid, making this approach impractical. While one can try to manually find a fixed $\mathcal{P}_B$ with a smaller expected trajectory length, this is generally a challenging problem, thus, we consider only the setting of trainable $\mathcal{P}_B$ here. Our findings are presented in Figure 2. Similarly to $7 \times 7$ grid, we find that learning in $\Delta \mathcal{F}$ scale results in a biased policy both for `DB` and `SDB`, and this bias is noticeably larger than in the smaller grid. In $\Delta \log \mathcal{F}$ scale, both `DB` and `SDB` employed with state flow regularization learn to correctly sample from the reward distribution. While all methods converge to similar expected trajectory length, $\Delta \mathcal{F}$ scale losses have smaller $n_\tau$ in the middle of the training even when employed without regularization, which supports our scaling hypothesis. In addition, Figure 4 in Appendix D

shows that for both losses in $\Delta \log \mathcal{F}$ scale, a mean length of sampled trajectories tends to infinity when the training is done without state flow regularization. Finally, we note that $\Delta \log \mathcal{F}$ losses correctly learn the true normalizing constant $\mathcal{Z}$, while $\Delta \mathcal{F}$ losses perform worse.

### 4.3. Permutations

Next, we consider the environment corresponding to the Cayley Graph of the symmetric group $S_n$ (group of permutations on $n$ elements $\{1, 2, \ldots, n\}$) from (Brunswic et al., 2024). Each state $s \in \mathcal{S} \setminus \{s_0, s_f\}$ is a permutation of fixed length $(s(1), \ldots, s(n))$, and there are $n - 1$ possible transitions that correspond to swapping a pair of adjacent elements $s(k)$ and $s(k+1)$, plus a transition that corresponds to a circular shift of the permutation to the right $(s(n), s(1), \ldots, s(n-1))$. In addition, each state has a terminating transition $s \to s_f$. GFlowNet reward utilized in the experiments of (Brunswic et al., 2024) is $\mathbb{I}[s(1) = 1]$. We argue that this results in a fairly simple task, and a trivial forward policy exists that just applies circular shift until 1 is in the first position. We opt for using a more complex reward distribution in our experiments and define GFlowNet reward in terms of the number of fixed points in a permutation $\mathcal{R}(s) = \exp\left(\frac{1}{2} \sum_{k=1}^{n} \mathbb{I}\{s(k) = k\}\right)$.

Since with the growth of $n$, the number of states $n!$ quickly becomes too large to compute total variation distance as it was done for hypergrid, we track convergence of a number of statistics to their true respective values. Firstly, we compute the relative error between the mean reward of GFlowNet samples and the true expected reward as it was proposed in (Shen et al., 2023). Secondly, denote $C(k)$ as the probability that a permutation sampled from the reward distribution has $k$ fixed points. We compute the $L^1$ error between the vector $(C(0), C(1), \ldots, C(n))$ and its empirical estimate over the last $10^5$ samples seen in training. Finally, we track the convergence of the trained $\log \mathcal{Z}_\theta$ to the true value of $\log \mathcal{Z}$. In Appendix C.3.1, we show how true reference values of these quantities can be efficiently computed.

Table 1 presents the results for $n = 8$ and $n = 20$. Here, $\mathcal{P}_B$ is trained in all cases. While for $n = 8$, the environment is still relatively small, $n = 20$ results in a more challenging environment with $\approx 2.4 \cdot 10^{18}$ states, thus the trained neural network needs to generalize to states unseen during training in order to match the reward distribution. Firstly, we note that while $\Delta \mathcal{F}$ scale losses can learn the reward distribution to some capacity for $n = 8$, they fail for $n = 20$. However, in all cases, they converge to small $\mathbb{E}[n_\tau]$, supporting our scaling hypothesis. On the other hand, we find that GFlowNets training with $\Delta \log \mathcal{F}$ losses and state flow regularization converges to low values of reward distribution approximation errors for both $n = 8$ and $n = 20$. In addi-

tion, we see that using a smaller regularization coefficient $\lambda$ on the one hand results in a model with a larger expected trajectory length, but on the other hand, results in a model that better matches the reward distribution. Finally, we perform the same experiment as for hypergrids (Figure 1) with a fixed $\mathcal{P}_B$ compared to a trainable $\mathcal{P}_B$ on small permutations of length $n = 4$, and make similar observations to the ones presented in Section 4.2. The results are presented in Figure 6 in Appendix D.

### 4.4. Discussion

The key observations from our experimental evaluation are:

> 1. Learning with a fixed $\mathcal{P}_B$ is possible without stable losses and regularization, however, manually picking $\mathcal{P}_B$ with small $\mathbb{E}[n_\tau]$ is challenging;
> 2. When $\mathcal{P}_B$ is trained, our results empirically support the scaling hypothesis, showing that even the standard DB in $\Delta \mathcal{F}$ scale is stable; however, non-acyclic GFlowNets trained with $\Delta \mathcal{F}$ scale losses often fail to accurately match the reward distribution;
> 3. Both DB and SDB in $\Delta \log \mathcal{F}$ scale result in better matching the reward distribution but need to be utilized with state flow regularization to ensure small expected trajectory length $\mathbb{E}[n_\tau]$.

## 5. Conclusion

In our paper we extended the theoretical framework of GFlowNets to encompass non-acyclic discrete environments, revisiting and simplifying the previous constructions by (Brunswic et al., 2024). In addition, we provided a number of theoretical insights regarding backward policies and the nature of flows in non-acyclic GFlowNets, generalized known connections between GFlowNets training and entropy-regularized RL to this setting, and experimentally re-examined the importance of the concept of loss stability proposed in (Brunswic et al., 2024).

Future work could explore applying other losses from acyclic GFlowNet literature (Madan et al., 2023; Silva et al., 2024; Hu et al., 2025) to the non-acyclic setting. Based on Theorem 3.13, another promising direction is to apply known RL techniques and algorithms to GFlowNets in the non-acyclic case, following their success for acyclic GFlowNets (Tiapkin et al., 2024; Mohammadpour et al., 2024; Lau et al., 2024; Morozov et al., 2024). Finally, environments where all states are terminal, i.e., have a transition into $s_f$, naturally arise in the non-acyclic case. Then, special modifications can be introduced to improve the propagation of the reward signal during training (Deleu et al., 2022; Pan et al., 2023; Jang et al., 2024b).

## Acknowledgements

We would like to thank Leo Maxime Brunswic for the helpful discussion and providing implementation details of the paper (Brunswic et al., 2024). This work was supported by the Ministry of Economic Development of the Russian Federation (code 25-139-66879-1-0003). This research was supported in part through computational resources of HPC facilities at HSE University (Kostenetskiy et al., 2021).

## Impact Statement

This paper presents work whose goal is to advance the field of Machine Learning. There are many potential societal consequences of our work, none which we feel must be specifically highlighted here.

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

# A. Proofs

## A.1. Proof of Lemma 3.4

Consider a random walk on $\mathcal{G}$ with reversed edges transition probabilities given by backward policy. Specifically, we define a Markov chain $\{X_n\}_{n=0}^{\infty}$, such that $X_0 = s_f$ a.s. and $\mathbb{P}[X_i = s \mid X_{i-1} = s'] = \mathcal{P}_{\mathrm{B}}(s \mid s')$. Let also $\mathbb{P}[X_i = s_0 \mid X_{i-1} = s_0] = 1$, i.e., $s_0$ is an absorbing state. We want to show that

1. The random walk terminates at $s_0$ with probability 1: $\mathbb{P}[\exists n : X_n = s_0] = 1$;

2. The expected length of a walk is finite: $\mathbb{E}[n_\tau] = \mathbb{E}[\sum_{n=0}^{\infty} \mathbb{I}\{X_i \neq s_0\}] < \infty$.

In particular, the first statement implies that $\mathcal{P}(\cdot)$ is a correct probability measure over finite trajectories since for any $\tau = (s_0, s_1 \ldots, s_{n_\tau}, s_f)$ it holds

$$\mathbb{P}[(X_{n_\tau+1}, \ldots, X_0) = \tau] = \prod_{t=0}^{n_\tau} \mathcal{P}_{\mathrm{B}}(s_t \mid s_{t+1}) = \mathcal{P}(\tau),$$

and we have

$$\sum_{\tau \in \mathcal{T}} \mathbb{P}[(X_{n_\tau+1}, \ldots, X_0) = \tau] = \mathbb{P}[\exists \tau \in \mathcal{T} : (X_{n_\tau+1}, \ldots, X_0) = \tau] = \mathbb{P}[\exists n : X_n = s_0],$$

since the events $\{(X_{n_\tau+1}, \ldots, X_0) = \tau\}$ do not intersect for different $\tau$.

First consider any intermediate state $s \in \mathcal{S} \setminus \{s_0, s_f\}$ and define a Markov chain $\{Y_n\}_{n=0}^{\infty}$ with the same transition probabilities as $\{X_n\}_{n=0}^{\infty}$, with $Y_0 = s$ a.s. We define $p_s \triangleq \mathbb{P}[\exists n > 0 : Y_n = s]$, i.e. the probability that $\{Y_n\}$ returns to $s$. First, notice that $p_s < 1$. Indeed, based on our assumptions on $\mathcal{G}$, there exists at least one path $\tau$ from $s_0$ to $s$, and furthermore, there exists such a path without cycles. In this case, we have

$$\mathbb{P}[(Y_{n_\tau+1}, \ldots, Y_0) = \tau] = \prod_{t=0}^{n_\tau} \mathcal{P}_{\mathrm{B}}(s_t \mid s_{t+1}) > 0$$

by the condition on $\mathcal{P}_{\mathrm{B}}(s \mid s') > 0$ for $s \to s' \in \mathcal{E}$. Notice that $\{(Y_{n_\tau+1}, \ldots, Y_0) = \tau\} \cap \{\exists n > 0 : Y_n = s\} = \emptyset$, since if the trajectory of the random walk has already reached $s_0$, it will never return to $s$. Thus, $p_s = \mathbb{P}[\exists n > 0 : Y_n = s] < 1$.

Next, for each state $s \in \mathcal{S} \setminus \{s_0, s_f\}$ we define $N_s \triangleq \sum_{i=0}^{\infty} \mathbb{I}\{X_i = s\}$ as the number of visits of a state $s$ encountered by the original process. Also, let us define $N_s' \triangleq \sum_{i=0}^{\infty} \mathbb{I}\{Y_i = s\}$ as the number of visits of a state $s$ during the backward random walk that starts at $s$. We notice that

$$\mathbb{E}[N_s] = \mathbb{P}(\exists k > 0 : X_k = s)\mathbb{E}[N_s'] \leq \mathbb{E}[N_s'],$$

where the first equation is due to the strong Markov property. At the same time, we have $\mathbb{P}[N_s' > k] = \mathbb{P}[\exists n_1, \ldots, n_k > 0 : Y_{n_j} = s]$ and, by Markov property, we have $\mathbb{P}[N_s' > k] = p_s^k$. Thus

$$\mathbb{E}[N_s'] = \sum_{k=0}^{\infty} \mathbb{P}[N_s' > k] = \sum_{k=0}^{\infty} p_s^k = \frac{1}{1 - p_s} < +\infty.$$

Finally, we have

$$\mathbb{E}[n_\tau] = \mathbb{E}\left[\sum_{k=0}^{\infty} \mathbb{I}\{X_k \neq s_0\}\right] = \mathbb{E}\left[\sum_{s \in \mathcal{S} \setminus \{s_0, s_f\}} \sum_{k=0}^{\infty} \mathbb{I}\{X_k = s\}\right] = \sum_{s \in \mathcal{S} \setminus \{s_0, s_f\}} \mathbb{E}[N_s] \leq \sum_{s \in \mathcal{S} \setminus \{s_0, s_f\}} \mathbb{E}[N_s'] < +\infty.$$

The first statement of the Lemma directly follows from the finiteness of the expected length of the walk, because otherwise it has an infinite length with non-zero probability, leading to a contradiction.

## A.2. Proof of Proposition 3.6

First, we prove the flow matching conditions. We have

$$\mathcal{F}(s \to s') = \mathcal{F}(s_f) \, \mathbb{E}_\tau \left[ \sum_{t=0}^{n_\tau} \mathbb{I}\{s_t = s, s_{t+1} = s'\} \right] = \mathcal{F}(s_f) \mathbb{E}_{n_\tau} \left[ \mathbb{E}_\tau \left[ \sum_{t=0}^{n_\tau+1} \mathbb{I}\{s_t = s, s_{t+1} = s'\} \mid n_\tau \right] \right],$$

$$\mathcal{F}(s) = \mathcal{F}(s_f) \, \mathbb{E}_\tau \left[ \sum_{t=0}^{n_\tau+1} \mathbb{I}\{s_t = s\} \right] = \mathcal{F}(s_f) \mathbb{E}_{n_\tau} \left[ \mathbb{E}_\tau \left[ \sum_{t=0}^{n_\tau+1} \mathbb{I}\{s_t = s\} \mid n_\tau \right] \right].$$

Next, note that the following equations hold for any trajectory $\tau$ and any $s \in \mathcal{S} \setminus \{s_0, s_f\}$:

$$\mathbb{I}\{s_t = s\} = \sum_{s'' \in \text{in}(s)} \mathbb{I}\{s_{t-1} = s'', s_t = s\} = \sum_{s' \in \text{out}(s)} \mathbb{I}\{s_t = s, s_{t+1} = s'\}.$$

Then for $s \in \mathcal{S} \setminus \{s_0\}$:

$$\mathcal{F}(s) = \mathcal{F}(s_f) \mathbb{E}_{n_\tau} \left[ \mathbb{E}_\tau \left[ \sum_{t=1}^{n_\tau+1} \sum_{s'' \in \text{in}(s)} \mathbb{I}\{s_{t-1} = s'', s_t = s\} \mid n_\tau \right] \right]$$

$$= \sum_{s'' \in \text{in}(s)} \mathcal{F}(s_f) \mathbb{E}_{n_\tau} \left[ \mathbb{E}_\tau \left[ \sum_{t=1}^{n_\tau+1} \mathbb{I}\{s_{t-1} = s'', s_t = s\} \mid n_\tau \right] \right] = \sum_{s'' \in \text{in}(s)} \mathcal{F}(s'' \to s).$$

Similarly for any $s \in \mathcal{S} \setminus \{s_f\}$:

$$\mathcal{F}(s) = \mathcal{F}(s_f) \mathbb{E}_{n_\tau} \left[ \mathbb{E}_\tau \left[ \sum_{t=0}^{n_\tau} \sum_{s' \in \text{out}(s)} \mathbb{I}\{s_t = s, s_{t+1} = s'\} \mid n_\tau \right] \right]$$

$$= \sum_{s' \in \text{out}(s)} \mathcal{F}(s_f) \mathbb{E}_{n_\tau} \left[ \mathbb{E}_\tau \left[ \sum_{t=0}^{n_\tau} \mathbb{I}\{s_t = s, s_{t+1} = s'\} \mid n_\tau \right] \right] = \sum_{s' \in \text{out}(s)} \mathcal{F}(s \to s').$$

Next, we prove the key relation $\mathcal{F}(s \to s') = \mathcal{F}(s')\mathcal{P}_\text{B}(s \mid s')$. We have

$$\mathcal{F}(s \to s') = \mathcal{F}(s_f) \mathbb{E}_{n_\tau} \left[ \mathbb{E}_\tau \left[ \sum_{t=0}^{n_\tau} \mathbb{I}\{s_t = s, s_{t+1} = s'\} \mid n_\tau \right] \right]$$

$$= \mathcal{F}(s_f) \mathbb{E}_{n_\tau} \left[ \sum_{t=0}^{n_\tau} \mathbb{E}_\tau [\mathbb{I}\{s_t = s, s_{t+1} = s'\} \mid n_\tau] \right]$$

$$= \mathcal{F}(s_f) \mathbb{E}_{n_\tau} \left[ \sum_{t=0}^{n_\tau} \mathbb{P}(s_t = s, s_{t+1} = s' \mid n_\tau) \right]$$

$$\overset{(a)}{=} \mathcal{F}(s_f) \mathbb{E}_{n_\tau} \left[ \sum_{t=0}^{n_\tau} \mathbb{P}(s_{t+1} = s' \mid n_\tau) \mathbb{P}(s_t = s \mid s_{t+1} = s', n_\tau) \right]$$

$$= \mathcal{F}(s_f) \mathbb{E}_{n_\tau} \left[ \sum_{t=0}^{n_\tau} \mathbb{P}(s_{t+1} = s' \mid n_\tau) \right] \mathcal{P}_\text{B}(s \mid s') = \mathcal{F}(s')\mathcal{P}_\text{B}(s \mid s').$$

Here in (a) we used the Markov property of trajectory distribution. Finally, by definition, $\mathcal{F}(s_0)$ is equal to, $\mathcal{F}(s_f)$ multiplied by the expected number of times $\tau \sim \mathcal{P}$ visits $s_0$, where the latter is always 1, so we have $\mathcal{F}(s_0) = \mathcal{F}(s_f)$.

## A.3. Proof of Proposition 3.7

Since $\mathcal{F}(s \to s')$ satisfies the flow matching conditions, we define

$$\mathcal{F}(s) = \sum_{s' \in \text{out}(s)} \mathcal{F}(s \to s') = \sum_{s'' \text{ in}(s)} \mathcal{F}(s'' \to s).$$

Next, take $\mathcal{P}_{\mathrm{B}}(s \mid s') = \mathcal{F}(s \to s')/\mathcal{F}(s')$. Let us denote $\hat{\mathcal{F}}$ to be flows from Definition 3.5 that are induced by $\mathcal{P}_{\mathrm{B}}$ and $\mathcal{F}(s_f)$ (which correspond to expected number of visits with respect to the trajectory distribution $\mathcal{P}$ induced by $\mathcal{P}_{\mathrm{B}}$). We aim to prove that $\mathcal{F}$ and $\hat{\mathcal{F}}$ coincide.

By Proposition 3.6 and definition of $\mathcal{P}_{\mathrm{B}}$, we have

$$\hat{\mathcal{F}}(s \to s') = \hat{\mathcal{F}}(s')\mathcal{P}_{\mathrm{B}}(s \mid s') = \frac{\hat{\mathcal{F}}(s')}{\mathcal{F}(s')}\mathcal{F}(s \to s') = C(s')\mathcal{F}(s \to s'),$$

where we denote $C(s) = \hat{\mathcal{F}}(s)/\mathcal{F}(s)$. In addition, by Proposition 3.6, $\hat{\mathcal{F}}$ satisfies the flow matching conditions, thus

$$\hat{\mathcal{F}}(s) = \sum_{s' \in \text{out}(s)} \hat{\mathcal{F}}(s \to s') = \sum_{s'' \text{ in}(s)} \hat{\mathcal{F}}(s'' \to s).$$

Combining these statements, for any $s \in \mathcal{S} \setminus \{s_f\}$ we have

$$\hat{\mathcal{F}}(s) = C(s)\mathcal{F}(s) = \sum_{s' \in \text{out}(s)} C(s')\mathcal{F}(s \to s').$$

The first equation is by definition of $C(s)$ and the second equation is due to the flow matching conditions. Thus we have a system of linear equations with respect to $C(s)$:

$$\forall s \in \mathcal{S} \setminus \{s_f\}, \quad \sum_{s' \in \text{out}(s)} C(s')\mathcal{F}(s \to s') - C(s)\mathcal{F}(s) = 0.$$

In addition, $\hat{\mathcal{F}}(s_f)$ is equal to, by definition, $\mathcal{F}(s_f)$ multiplied by the expected number of times $\tau \sim \mathcal{P}$ visits $s_f$, where the latter is 1, so we have $\hat{\mathcal{F}}(s_f) = \mathcal{F}(s_f)$, thus an additional equation is $C(s_f) = 1$. In total, we have $|\mathcal{S}|$ variables and $|\mathcal{S}|$ equations, and are interested in strictly positive solutions. Firstly, there exists a trivial solution $C(s) = 1$ for each $s \in \mathcal{S}$, which is an only constant solution since $C(s_f) = 1$.

Next, suppose there exists a non-constant solution $C'(s)$. Denote $\mathcal{S}_{\max} = \text{argmax}_{s \in \mathcal{S}} C'(s)$, which will be a proper subset of $\mathcal{S}$. Let us consider two cases. First, suppose $s_f \notin \mathcal{S}_{\max}$. Let $\tau = (s_0 \to s_1 \to \ldots \to s_{n_\tau} \to s_f)$ be any trajectory that visits some state in $\mathcal{S}_{\max}$. Then there exists an index $t \leq n_\tau$ such that $s_t \in \mathcal{S}_{\max}$ and $s_{t+1} \notin \mathcal{S}_{\max}$. Then we have

$$1 = \frac{\sum_{s' \in \text{out}(s_t)} C(s')\mathcal{F}(s_t \to s')}{C(s_t)\mathcal{F}(s_t)} < \frac{\sum_{s' \in \text{out}(s_t)} C(s_t)\mathcal{F}(s_t \to s')}{C(s_t)\mathcal{F}(s_t)} = \frac{\sum_{s' \in \text{out}(s_t)} \mathcal{F}(s_t \to s')}{\mathcal{F}(s_t)} = 1.$$

The inequality is due to three facts: (i) $C(s) > 0 \ \forall s \in \mathcal{S}$, (ii) $s_t \in \mathcal{S}_{\max}$, and thus $C(s_t) \geq C(s')$ for any $s' \in \mathcal{S}$, and (iii) the inequality is strict for at least one edge $s_t \to s_{t+1}$ such that $C'(s_{t+1}) < C'(s_t)$, and it implies a contradiction.

Second, suppose $s_f \in \mathcal{S}_{\max}$. Then, denote $\mathcal{S}_{\min} = \text{argmin}_{s \in \mathcal{S}} C'(s)$, which will be a proper subset of $\mathcal{S}$. Similarly to the previous case, let $\tau$ be any trajectory that visits some state in $\mathcal{S}_{\min}$. Then there exists an index $t \leq n_\tau$ such that $s_t \in \mathcal{S}_{\min}$ and $s_{t+1} \notin \mathcal{S}_{\min}$. Then we have

$$1 = \frac{\sum_{s' \in \text{out}(s_t)} C(s')\mathcal{F}(s_t \to s')}{C(s_t)\mathcal{F}(s_t)} > \frac{\sum_{s' \in \text{out}(s_t)} C(s_t)\mathcal{F}(s_t \to s')}{C(s_t)\mathcal{F}(s_t)} = \frac{\sum_{s' \in \text{out}(s_t)} \mathcal{F}(s_t \to s')}{\mathcal{F}(s_t)} = 1.$$

Thus, in this case, there is also a contradiction. Therefore $C(s) = 1$ is a unique solution, meaning that $\hat{\mathcal{F}}(s) = \mathcal{F}(s)$. Finally for any $s \to s' \in \mathcal{E}$

$$\hat{\mathcal{F}}(s \to s') = C(s')\mathcal{F}(s \to s') = \mathcal{F}(s \to s').$$

$\mathcal{F}$ and $\hat{\mathcal{F}}$ coincide, thus the proposition is proven.

### A.4. Proof of Proposition 3.8

Let us proof existence and uniqueness of a corresponding forward policy. Let $\mathcal{F}$ be the flow induced by the backward policy (3.5).

**Uniqueness.** Suppose that such a forward policy $\mathcal{P}_F$ exists, then probability distributions over $\mathcal{T}$ induced by $\mathcal{P}_B$ and $\mathcal{P}_F$ coincide. Then

$$
\begin{aligned}
\mathcal{F}(s \to s') &= \mathcal{F}(s_f)\mathbb{E}_{n_\tau}\left[\mathbb{E}_\tau\left[\sum_{t=0}^{n_\tau}\mathbb{I}\{s_t = s, s_{t+1} = s'\} \mid n_\tau\right]\right] \\
&= \mathcal{F}(s_f)\mathbb{E}_{n_\tau}\left[\sum_{t=0}^{n_\tau}\mathbb{E}_\tau[\mathbb{I}\{s_t = s, s_{t+1} = s'\} \mid n_\tau]\right] \\
&= \mathcal{F}(s_f)\mathbb{E}_{n_\tau}\left[\sum_{t=0}^{n_\tau}\mathbb{P}(s_t = s, s_{t+1} = s' \mid n_\tau)\right] \\
&= \mathcal{F}(s_f)\mathbb{E}_{n_\tau}\left[\sum_{t=0}^{n_\tau}\mathbb{P}(s_t = s \mid n_\tau)\mathbb{P}(s_{t+1} = s' \mid s_t = s, n_\tau)\right] \\
&= \mathcal{F}(s_f)\mathbb{E}_{n_\tau}\left[\sum_{t=0}^{n_\tau}\mathbb{P}(s_t = s \mid n_\tau)\right]\mathcal{P}_F(s' \mid s) = \mathcal{F}(s)\mathcal{P}_F(s' \mid s).
\end{aligned}
$$

Then we have $\mathcal{F}(s')\mathcal{P}_B(s \mid s') = \mathcal{F}(s)\mathcal{P}_F(s' \mid s)$ and $\mathcal{P}_F(s' \mid s) = \mathcal{F}(s')\mathcal{P}_B(s \mid s')/\mathcal{F}(s)$, thus we finish the proof of uniqueness presented above.

**Existence.** Take

$$
\mathcal{P}_F(s' \mid s) = \frac{\mathcal{F}(s')\mathcal{P}_B(s \mid s')}{\mathcal{F}(s)} = \frac{\mathcal{F}(s \to s')}{\mathcal{F}(s)}.
$$

This is always a valid probability distribution since $\mathcal{F}(s) = \sum_{s' \in \text{out}(s)}\mathcal{F}(s \to s')$. Next, for any $\tau \in \mathcal{T}$ we have

$$
\prod_{t=0}^{n_\tau}\mathcal{P}_B(s_t \mid s_{t+1}) = \prod_{t=0}^{n_\tau}\frac{\mathcal{F}(s_t \to s_{t+1})}{\mathcal{F}(s_{t+1})} = \frac{\prod_{t=0}^{n_\tau}\mathcal{F}(s_t \to s_{t+1})}{\prod_{t=0}^{n_\tau}\mathcal{F}(s_{t+1})}.
$$

where the first equation is due to Proposition 3.6. Next, note that

$$
\mathcal{F}(s_0) = \mathcal{F}(s_f)\,\mathbb{E}_\tau\left[\sum_{t=0}^{n_\tau+1}\mathbb{I}\{s_t = s_0\}\right] = \mathcal{F}(s_f) \cdot 1 = \mathcal{F}(s_f).
$$

Then

$$
\frac{\prod_{t=0}^{n_\tau}\mathcal{F}(s_t \to s_{t+1})}{\prod_{t=0}^{n_\tau}\mathcal{F}(s_{t+1})} = \frac{\mathcal{F}(s_0)}{\mathcal{F}(s_f)}\frac{\prod_{t=0}^{n_\tau}\mathcal{F}(s_t \to s_{t+1})}{\prod_{t=0}^{n_\tau}\mathcal{F}(s_t)} = \frac{\prod_{t=0}^{n_\tau}\mathcal{F}(s_t \to s_{t+1})}{\prod_{t=0}^{n_\tau}\mathcal{F}(s_t)} = \prod_{t=0}^{n_\tau}\mathcal{P}_F(s_{t+1} \mid s_t).
$$

Thus the existence is proven. Finally, the detailed balance conditions follow from the proof of uniqueness.

### A.5. Proof of Proposition 3.9

Consider an edge function $\bar{\mathcal{F}}(s \to s') = \mathcal{F}(s)\mathcal{P}_F(s' \mid s)$. It is positive since $\mathcal{F}(s) > 0$ and $\mathcal{P}_F(s' \mid s) > 0$ by the statement of the proposition. Since $\mathcal{P}_F(\cdot \mid s)$ is a valid probability distribution over $\text{out}(s)$, we have

$$
\sum_{s' \in \text{out}(s)}\bar{\mathcal{F}}(s \to s') = \sum_{s' \in \text{out}(s)}\mathcal{F}(s)\mathcal{P}_F(s' \mid s) = \mathcal{F}(s)\sum_{s' \in \text{out}(s)}\mathcal{P}_F(s' \mid s) = \mathcal{F}(s).
$$

Similarly, since $\mathcal{P}_B(\cdot \mid s)$ is a valid probability distribution over $\text{in}(s)$, and $\mathcal{F}$, $\mathcal{P}_F$ and $\mathcal{P}_B$ satisfy the detailed balance conditions, we have

$$
\sum_{s'' \in \text{in}(s)}\bar{\mathcal{F}}(s'' \to s) = \sum_{s'' \in \text{in}(s)}\mathcal{F}(s'')\mathcal{P}_F(s \mid s'') = \sum_{s'' \in \text{in}(s)}\mathcal{F}(s)\mathcal{P}_B(s'' \mid s) = \mathcal{F}(s)\sum_{s'' \in \text{in}(s)}\mathcal{P}_B(s'' \mid s) = \mathcal{F}(s).
$$

Thus $\bar{\mathcal{F}}$ satisfies the flow matching conditions. In addition

$$
\mathcal{P}_B(s \mid s') = \frac{\mathcal{F}(s)\mathcal{P}_F(s' \mid s)}{\mathcal{F}(s')} = \frac{\bar{\mathcal{F}}(s \to s')}{\mathcal{F}(s')} = \frac{\bar{\mathcal{F}}(s \to s')}{\sum_{s'' \in \text{in}(s')}\bar{\mathcal{F}}(s'' \to s')}.
$$

Thus, applying Proposition 3.7 to $\bar{\mathcal{F}}$, we get that it is an edge flow induced by $\mathcal{P}_{\mathrm{B}}$, thus $\mathcal{F}$ is also the state flow induced by $\mathcal{P}_{\mathrm{B}}$ and $\mathcal{F}(s_f)$.

Next, consider any trajectory $\tau = (s_0, s_1, \ldots, s_{n_\tau}, s_f) \in \mathcal{T}$. By the detailed balance conditions we have

$$\prod_{t=0}^{n_\tau} \mathcal{P}_{\mathrm{B}}(s_t \mid s_{t+1}) = \prod_{t=0}^{n_\tau} \frac{\mathcal{F}(s_t)\mathcal{P}_{\mathrm{F}}(s_{t+1} \mid s_t)}{\mathcal{F}(s_{t+1})} = \frac{\mathcal{F}(s_0)}{\mathcal{F}(s_f)} \prod_{t=0}^{n_\tau} \mathcal{P}_{\mathrm{F}}(s_{t+1} \mid s_t) = \prod_{t=0}^{n_\tau} \mathcal{P}_{\mathrm{F}}(s_{t+1} \mid s_t).$$

The final equation is due to the fact that state flow $\mathcal{F}$ is induced by $\mathcal{P}_{\mathrm{B}}$ and $\mathcal{F}(s_f)$, so we have $\mathcal{F}(s_f) = \mathcal{F}(s_0)$ by Proposition 3.6. Thus the proposition is proven.

### A.6. Proof of Proposition 3.12

We first note that not including $s_0$ and $s_f$ in the sum is just a matter of the definition of trajectory length presented in 2.1, where we do not count the first and the final state towards it. Using the fact that the length of a trajectory is the sum of the numbers of visits to each individual state in the graph, we obtain that

$$\mathbb{E}_{\tau \sim \mathcal{P}}[n_\tau] = \mathbb{E}_\tau\left[\sum_{s \in \mathcal{S}\setminus\{s_0, s_f\}} \sum_{t=0}^{n_\tau+1} \mathbb{I}\{s_t = s\}\right] = \sum_{s \in \mathcal{S}\setminus\{s_0, s_f\}} \mathbb{E}_\tau\left[\sum_{t=0}^{n_\tau+1} \mathbb{I}\{s_t = s\}\right] = \sum_{s \in \mathcal{S}\setminus\{s_0, s_f\}} \frac{\mathcal{F}(s)}{\mathcal{F}(s_f)}.$$

### A.7. Entropy-Regularized RL and Theorem 3.13

**Background on Entropy-Regularized RL.** Let $\mathcal{M}_\mathcal{G}$ be a deterministic MDP induced by a graph $\mathcal{G}$ with a state space $\mathcal{S}$ corresponding to vertices of $\mathcal{G}$, the action space $\mathcal{A}_s$ for each state $s$ corresponds to outgoing edges of $s$, associated with $\mathrm{out}(s)$, and let $\lambda > 0$ be a regularization coefficient. We define a policy $\pi$ as a mapping from each state $s \in \mathcal{S}$ to a probability measure $\pi(\cdot|s)$ over $\mathcal{A}_s$.

Then, for any policy $\pi$, we define the regularized value function for all $s \neq s_f$ as follows

$$V_\lambda^\pi(s) \triangleq \mathbb{E}_{\tau \sim \mathsf{P}_\mathcal{T}^\pi}\left[\sum_{t=0}^{n_\tau} r(s_t, s_{t+1}) + \lambda\mathcal{H}(\pi(\cdot|s_t)) \mid s_0 = s\right], \tag{13}$$

and $V_\lambda^\pi(s_f) = 0$, where $\mathcal{H}$ is Shannon entropy, $\mathsf{P}_\mathcal{T}^\pi$ is a trajectory distribution induced by the following the policy $\pi$: $s_t \sim \pi(\cdot|s_{t-1})$ for all $t \geq 1$ and the starting state $s_0$ is fixed as $s$ (not to be confused with the initial state in $\mathcal{G}$), and $n_\tau$ is a length of trajectory defined as $n_\tau = \min\{k \geq 0 \mid s_{k+1} = s_f\}$. Overall, it is not clear if the value function is a well-defined function when no discounting is used ($\gamma = 1$). We call this problem a *regularized shortest path* problem, akin to shortest path and stochastic shortest path problem (Bertsekas, 2012, Chapter 3). A policy $\pi^\star$ is called optimal if it maximizes $V_\lambda^\pi(s_0)$.

**Lemma A.1.** *Assume that (i) a graph $\mathcal{G}$ satisfies Assumption 3.1 and (ii) for any $s \in \mathcal{S}, s' \in \mathcal{A}_s$ it holds $r(s, s') \leq 0$ and $r(s, s') = 0$ if and only if $|\mathrm{in}(s')| = 1$. Also, assume that for any optimal policy $\pi^\star$ it holds $\mathbb{E}_{\pi^\star}[n_\tau] < +\infty$.*

*Then, a regularized shortest path problem admits a unique solution, and the value of its solution satisfies soft optimal Bellman equations*

$$Q_\lambda^\star(s, s') \triangleq r(s, s') + V^\star(s'), \qquad V_\lambda^\star(s) \triangleq \lambda \log\left(\sum_{s' \in \mathrm{out}(s)} \exp\left\{\frac{1}{\lambda}Q_\lambda^\star(s, s')\right\}\right), \tag{14}$$

*where the optimal policy can be derived as $\pi^\star(a|s) \propto \exp\{1/\lambda \cdot Q_\lambda^\star(s, a)\}$.*

*Proof.* Let us define a number of visits of a vertex $s$ and an edge $s, s'$ in $\mathcal{G}$ on a given trajectory $\tau$ as $n_\tau(s) = \sum_{t=0}^\infty \mathbb{I}\{s_t = s\}$ and $n_\tau(s, s') = \sum_{t=0}^\infty \mathbb{I}\{s_t = s, s_{t+1} = s'\}$. In the analogy with occupancy measures in RL, we employ the notation $d^\pi(s) \triangleq \mathbb{E}_\pi[n_\tau(s)]$ and $d^\pi(s, s') \triangleq \mathbb{E}_\pi[n_\tau(s, s')]$ for an expected number of visits. This definition also corresponds to the flow function in the reversed graph (see Definition 3.5) with the "backward policy" $\pi$. The condition on expected trajectory

length of optimal policies implies that we can consider only policies $\pi$ such that $d^\pi(s) < +\infty$ for any $s \in \mathcal{S} \setminus \{s_0, s_f\}$, and, as a result, $d^\pi(s, s') < +\infty$.

Next, we rewrite the value function in the initial state as follows

$$V_\lambda^\pi(s_0) = \sum_{s \in \mathcal{S} \setminus \{s_f\}} \sum_{s' \in \text{out}(s)} d^\pi(s, s') r(s, s') + \lambda \sum_{s \in \mathcal{S}} d^\pi(s) \mathcal{H}(\pi(\cdot|s)) \,.$$

Then, we notice that $d^\pi(s, s') = \pi(s'|s) \cdot d^\pi(s)$ thus we can rewrite the value in the following form

$$V_\lambda^\pi(s_0) = \sum_{s \in \mathcal{S} \setminus \{s_f\}} \sum_{s' \in \text{out}(s)} d^\pi(s, s') r(s, s') - \lambda \underbrace{\sum_{s \in \mathcal{S}} \sum_{s' \in \text{out}(s)} d^\pi(s, s') \log \left( d^\pi(s, s') / \sum_{s' \in \text{out}(s)} d^\pi(s, s') \right)}_{R(d^\pi)} \,.$$

As a function of $d^\pi(s, s')$, we see that the first term in the expression above is linear whereas the second one is relative conditional entropy (Neu et al., 2017) that is strongly concave. Given that the set of all admissible $d^\pi(s, s')$ is a polytope that is defined as a family of negative functions that satisfies the flow matching conditions (see Proposition 3.6)

$$\mathcal{K} \triangleq \left\{ d \colon \mathcal{S} \times \mathcal{S} \to \mathbb{R}_+ \,\middle|\, \sum_{s' \in \text{out}(s)} d^\pi(s, s') = \sum_{s'' \in \text{in}(s)} d^\pi(s'', s), \sum_{s' \in \text{out}(s_0)} d(s_0, s') = 1, \sum_{s'' \in \text{in}(s_f)} d(s'', s_f) = 1 \right\},$$

where the flow and policy have one-to-one corresponds due to Proposition 3.7 in the reversed graph. Since the set $\mathcal{K}$ is a polytope, optimization of $V_\lambda^\pi(s_0)$ over occupancy measures is a strictly convex problem and thus admits a unique solution $d^\star$ that corresponds to a unique policy $\pi^\star$.

Before proving the optimal Bellman equations, we want to show that $\pi^\star$ satisfies $\pi^\star(s'|s) > 0$ for any $s \in \mathcal{S}, s' \in \text{out}(s)$. To do it, we explore the gradients of the regularizer, using computations done in (Neu et al., 2017), Section A.4: $\frac{\partial R(d^\pi)}{\partial d^\pi(s, s')} = \log \pi(s'|s)$. In particular, it implies that as $\pi(s'|s) \to 0$, then $\|\nabla_{d^\pi} \partial R(d^\pi)\| \to +\infty$, thus the optimal policy $\pi^\star$ cannot satisfy $\pi^\star(s'|s) = 0$ since it will violate the optimality conditions.

Next, we want to prove that the value of the optimal policy satisfies soft optimal Bellman equations. First, we notice that the usual Bellman equations still hold since $n_\tau$ is a stopping time

$$Q_\lambda^\pi(s, s') = r(s, s') + V_\lambda^\pi(s'), \qquad V_\lambda^\pi(s) = \sum_{s' \in \text{out}(s)} \pi(s'|s) Q_\lambda^\pi(s, s') + \lambda \mathcal{H}(\pi(\cdot|s)) \,,$$

with an additional initial condition $V_\lambda^\pi(s_f) = 0$, by the proprieties of conditional expectation. Let us consider a regularized policy improvement operation, defined as

$$\pi'(\cdot|s) \triangleq \arg\max_p \left\{ \sum_{s' \in \text{out}(s)} p(s') Q_\lambda^\pi(s, s') + \lambda \mathcal{H}(p) \right\} \propto \exp\left\{ \frac{1}{\lambda} Q_\lambda^\pi(s, \cdot) \right\} \,.$$

Then we want to show that $V_\lambda^{\pi'}(s_0) \geq V_\lambda^\pi(s_0)$ if the policy $\pi$ is positive: $\pi(s'|s) > 0$ for all $s \in \mathcal{S}, s' \in \text{out}(s)$. We start from a general inequality that holds for any $s \in \mathcal{S}$

$$
\begin{aligned}
V_\lambda^{\pi'}(s) - V_\lambda^\pi(s) &= \left( \sum_{s' \in \text{out}(s)} \pi'(s'|s) Q_\lambda^{\pi'}(s, s') + \lambda \mathcal{H}(\pi'(\cdot|s)) \right) - \left( \sum_{s' \in \text{out}(s)} \pi(s'|s) Q_\lambda^\pi(s, s') + \lambda \mathcal{H}(\pi(\cdot|s)) \right) \\
&= \left( \sum_{s' \in \text{out}(s)} \pi'(s'|s) Q_\lambda^{\pi'}(s, s') + \lambda \mathcal{H}(\pi'(\cdot|s)) \right) - \left( \sum_{s' \in \text{out}(s)} \pi'(s'|s) Q_\lambda^\pi(s, s') + \lambda \mathcal{H}(\pi'(\cdot|s)) \right) \\
&\quad + \left( \sum_{s' \in \text{out}(s)} \pi'(s'|s) Q_\lambda^\pi(s, s') + \lambda \mathcal{H}(\pi'(\cdot|s)) \right) - \left( \sum_{s' \in \text{out}(s)} \pi(s'|s) Q_\lambda^\pi(s, s') + \lambda \mathcal{H}(\pi(\cdot|s)) \right) \\
&\geq \sum_{s' \in \text{out}(s)} \pi'(s'|s) \left[ Q_\lambda^{\pi'}(s, s') - Q_\lambda^\pi(s, s') \right] = \sum_{s' \in \text{out}(s)} \pi'(s'|s) \left[ V_\lambda^{\pi'}(s') - V_\lambda^\pi(s') \right] \,.
\end{aligned}
$$

After $t$ rollouts, we have

$$V_\lambda^{\pi'}(s_0) - V_\lambda^\pi(s_0) \geq \mathbb{E}_{\tau \sim \mathsf{P}_{\mathcal{T}}^{\pi'}}\left[V_\lambda^{\pi'}(s_t) - V_\lambda^\pi(s_t)\right],$$

Since the policy $\pi$ is positive, then Lemma 3.4 in the reversed graph implies that $d^\pi(s, s'), d^\pi(s) < +\infty$ and thus all values and Q-values are finite: $Q^\pi(s, s') > -\infty$ for any $s \in \mathcal{S}, s' \in \text{out}(s)$. It implies that $\pi'$ is also positive. Thus, its trajectories are finite with probability 1 and yields $V_\lambda^{\pi'}(s_0) \geq V_\lambda^\pi(s_0)$. Finally, applying policy improvement to $\pi^\star$ we conclude the statement. $\qquad\square$

*Proof of Theorem 3.13.* Let $\mathcal{P}$ be the trajectory distribution induced by the GFlowNet backward policy and $\mathsf{P}_{\mathcal{T}}^\pi$ be the trajectory distribution induced by RL policy $\pi$. Then we rewrite the value function (13) in the following form using the tower property of conditional expectation to replace entropy with negative logarithm of the policy

$$V_{\lambda=1}^\pi(s_0) = \mathbb{E}_{\tau \sim \mathsf{P}_{\mathcal{T}}^\pi}\left[\sum_{t=0}^{n_\tau} r(s_t, s_{t+1}) - \log \pi(s_{t+1} \mid s_t)\right].$$

Notice that there is no coefficient in front of entropy and reward because we set $\gamma = 1, \lambda = 1$ by the theorem statement. Using simple algebraic manipulations

$$V_{\lambda=1}^\pi(s_0) = \mathbb{E}_{\tau \sim \mathsf{P}_{\mathcal{T}}^\pi}\left[\sum_{t=0}^{n_\tau} \log \exp(r(s_t, s_{t+1})) - \log \pi(s_{t+1} \mid s_t)\right] = \mathbb{E}_{\tau \sim \mathsf{P}_{\mathcal{T}}^\pi}\left[\log \frac{\prod_{t=0}^{n_\tau} \exp(r(s_t, s_{t+1}))}{\prod_{t=0}^{n_\tau} \pi(s_{t+1} \mid s_t)}\right].$$

Next, we notice that $r(s, s') = \log \mathcal{P}_{\mathrm{B}}(s|s')$ for all non-terminal $s'$ and, $r(s, s_f) = \log \mathcal{R}(s) = \log \mathcal{P}_{\mathrm{B}}(s \mid s_f) + \log \mathcal{Z}$ for terminal transitions due to the reward matching condition. Thus,

$$V_{\lambda=1}^\pi(s_0) = \log \mathcal{Z} - \mathbb{E}_{\tau \sim \mathsf{P}_{\mathcal{T}}^\pi}\left[\log \frac{\prod_{t=0}^{n_\tau} \pi(s_{t+1} \mid s_t)}{\prod_{t=0}^{n_\tau} P_B(s_t \mid s_{t+1})}\right] = \log \mathcal{Z} - \mathrm{KL}(\mathsf{P}_{\mathcal{T}}^\pi | \mathcal{P}).$$

Here $\mathcal{P}$ is a trajectory distribution induced by $\mathcal{P}_{\mathrm{B}}$ (7). We note that the final equation is the same as the one in Proposition 1 of (Tiapkin et al., 2024) for the acyclic case.

Thus, an optimal policy $\pi^\star$ that maximizes $V_{\lambda=1}^\pi(s_0)$ is the one that minimizes $\mathrm{KL}(\mathsf{P}_{\mathcal{T}}^\pi | \mathcal{P})$. By Lemma 3.4, the expected trajectory length of any optimal policy $\pi^\star$ that matches $\mathcal{P}$ is finite. Thus, by Proposition 3.8, there exists a unique forward policy $\mathcal{P}_{\mathrm{F}}$ that induces the same trajectory distribution as $\mathcal{P}_{\mathrm{B}}$, which is equivalent to achieving zero KL-divergence. Thus, $\pi^\star$ coincides with $\mathcal{P}_{\mathrm{F}}$, and we conclude the statement by the uniqueness of the solution (see Lemma A.1). To apply Lemma A.1, without loss of generality, we can assume that the GFlowNet reward function $\mathcal{R}$ is normalized, i.e., $\mathcal{Z} = 1$ and $\log \mathcal{R}(x) < 0$. Indeed, since a terminating transition $x \to s_f$ is always visited exactly once, it is equivalent to subtracting $\log \mathcal{Z}$ from all terminal rewards, which does not change the optimal policy and modifies all values by the same constant.

Next, consider soft optimal Bellman equations (14) for non-terminating transitions

$$Q_{\lambda=1}^\star(s, s') = \log \mathcal{P}_{\mathrm{B}}(s \mid s') + \log \sum_{s'' \in \text{out}(s')} \exp(Q_{\lambda=1}^\star(s', s'')).$$

Let us show that $Q_{\lambda=1}^\star(s, s') = \log \mathcal{F}(s \to s')$ will satisfy the equations.

$$\log \mathcal{F}(s \to s') = \log \mathcal{F}(s') + \log \mathcal{P}_{\mathrm{B}}(s \mid s') = \log \sum_{s'' \in \text{out}(s')} \mathcal{F}(s' \to s'') + \log \mathcal{P}_{\mathrm{B}}(s \mid s')$$

$$= \log \mathcal{P}_{\mathrm{B}}(s \mid s') + \log \sum_{s'' \in \text{out}(s')} \exp(\log \mathcal{F}(s' \to s'')).$$

Here we used equations from Proposition 3.6. For terminating transitions we simply have $Q_{\lambda=1}^\star(s, s_f) = r(s, s_f) = \log \mathcal{R}(s) = \log \mathcal{F}(s \to s_f)$. Since there exists a unique solution to soft optimal Bellman equations, we have proven $Q_{\lambda=1}^\star(s, s') = \log \mathcal{F}(s \to s')$. As for state flows, we have

$$V_{\lambda=1}^\star(s) = \log \sum_{s' \in \text{out}(s)} \exp(Q_{\lambda=1}^\star(s, s')) = \log \sum_{s' \in \text{out}(s)} \exp(\log \mathcal{F}(s \to s')) = \log \mathcal{F}(s).$$

Thus the proof is concluded. $\qquad\square$

# B. Algorithmic Details

## B.1. Training Policy and Flow Weighting

Recall the optimization problem in (11):

$$\min_{\mathcal{F},\mathcal{P}_\mathrm{F},\mathcal{P}_\mathrm{B}} \sum_{s \in \mathcal{S} \setminus \{s_0, s_f\}} \mathcal{F}(s)$$

$$\text{subject to } \left( \log \frac{\mathcal{F}(s)\mathcal{P}_\mathrm{F}(s' \mid s)}{\mathcal{F}(s')\mathcal{P}_\mathrm{B}(s \mid s')} \right)^2 = 0 \,, \qquad\qquad \forall s \to s' \in \mathcal{E} \,,$$

$$\mathcal{F}(s_f)\mathcal{P}_\mathrm{B}(x|s_f) = \mathcal{R}(x) \,, \qquad\qquad \forall x \to s_f \in \mathcal{E} \,.$$

Now, suppose that training with DB loss (4) and state flow regularization (12) is done on-policy, i.e. trajectories are collected using the trained policy $\mathcal{P}_\mathrm{F}$. Let us write down the expected gradient of the loss summed over a trajectory (note that regularization is not applied to $\mathcal{F}(s_0)$ and $\mathcal{F}(s_f)$)

$$\mathbb{E}_{\tau \sim \mathcal{P}_\mathrm{F}} \left[ \sum_{t=0}^{n_\tau} \nabla_\theta \left( \log \frac{\mathcal{F}_\theta(s)\mathcal{P}_\mathrm{F}(s_{t+1} \mid s_t, \theta)}{\mathcal{F}_\theta(s_{t+1})\mathcal{P}_\mathrm{B}(s_t \mid s_{t+1}, \theta)} \right)^2 + \sum_{t=1}^{n_\tau} \lambda \nabla_\theta \mathcal{F}_\theta(t) \right],$$

which can be rewritten as

$$\mathbb{E}_{\tau \sim \mathcal{P}_\mathrm{F}} \left[ \sum_{t=0}^{n_\tau} \nabla_\theta \mathcal{L}_\mathrm{DB}(s_t \to s_{t+1}) \right] + \lambda \mathbb{E}_{\tau \sim \mathcal{P}_\mathrm{F}} \left[ \sum_{t=1}^{n_\tau} \nabla_\theta \mathcal{F}_\theta(s_t) \right].$$

The first term is the expected gradient of the standard DB loss. As for the second term, we note that if $\mathcal{F}_\theta$ is exactly the state flow induced by $\mathcal{P}_\mathrm{F}$, we have

$$\mathbb{E}_{\tau \sim \mathcal{P}_\mathrm{F}} \left[ \sum_{t=1}^{n_\tau} \nabla_\theta \mathcal{F}_\theta(s_t) \right] = \mathbb{E}_{\tau \sim \mathcal{P}_\mathrm{F}} \left[ \sum_{s \in \mathcal{S} \setminus \{s_0, s_f\}} \sum_{t=0}^{n_\tau} \mathbb{I}\{s_t = s\} \nabla_\theta \mathcal{F}_\theta(s) \right]$$

$$= \sum_{s \in \mathcal{S} \setminus \{s_0, s_f\}} \nabla_\theta \mathcal{F}_\theta(s) \mathbb{E}_{\tau \sim \mathcal{P}_\mathrm{F}} \left[ \sum_{t=0}^{n_\tau} \mathbb{I}\{s_t = s\} \right]$$

$$= \sum_{s \in \mathcal{S} \setminus \{s_0, s_f\}} \frac{\mathcal{F}_\theta(s)}{\mathcal{F}_\theta(s_f)} \nabla_\theta \mathcal{F}_\theta(s) = \frac{1}{2\mathcal{F}_\theta(s_f)} \nabla_\theta \left( \sum_{s \in \mathcal{S} \setminus \{s_0, s_f\}} \mathcal{F}_\theta(s)^2 \right).$$

This implies that on-policy training tries to minimize the sum of squared state flows rather than the sum of state flows. This happens due to the fact that the trajectory distribution that is used to collect data for training (induced by $\mathcal{P}_\mathrm{F}$ in this case) visits certain states more often than others, thus a weight is given to the flow in each state equal to the expected number of visits. However, if $\mathcal{P}_\mathrm{F}(s \mid s_0)$ is fixed to be uniform over $\mathcal{S} \setminus \{s_0, s_f\}$ (see Section 4 and Appendix B.3), this issue can be circumvented by applying the flow regularizer only in the first state of each sampled trajectory. Then, equal weight will be given to $\mathcal{F}_\theta(s)$ in each state in the expected loss, thus we will be minimizing the sum of state flows. However, in our experiments we noticed that this does not significantly influence the results, so we leave exploring this phenomenon as a further research direction.

## B.2. Loss Scaling and Stability

In this section, we provide a more detailed explanation of our scaling hypothesis (see Section 4). Let us consider a GFlowNet that learns $\mathcal{F}$, $\mathcal{P}_\mathrm{F}$ and $\mathcal{P}_\mathrm{B}$. Since these quantities are predicted by a neural network, a standard way is to make it predict logits for the forward policy, logits for the backward policy, and the logarithm of the state flow. Flow functions are always positive, thus, predicting them in log scale is a natural approach (Bengio et al., 2021; 2023). Then, for any transition $s \to s'$, define two quantities:

$$\Delta_{\log \mathcal{F}}(s, s', \theta) \triangleq \log \mathcal{F}_\theta(s) + \log \mathcal{P}_\mathrm{F}(s'|s, \theta) - \log \mathcal{F}_\theta(s') - \log \mathcal{P}_\mathrm{B}(s|s', \theta) \,,$$
$$\Delta_{\mathcal{F}}(s, s', \theta) \triangleq \exp(\log \mathcal{F}_\theta(s) + \log \mathcal{P}_\mathrm{F}(s'|s, \theta)) - \exp(\log \mathcal{F}_\theta(s') + \log \mathcal{P}_\mathrm{B}(s|s', \theta)) \,. \tag{15}$$

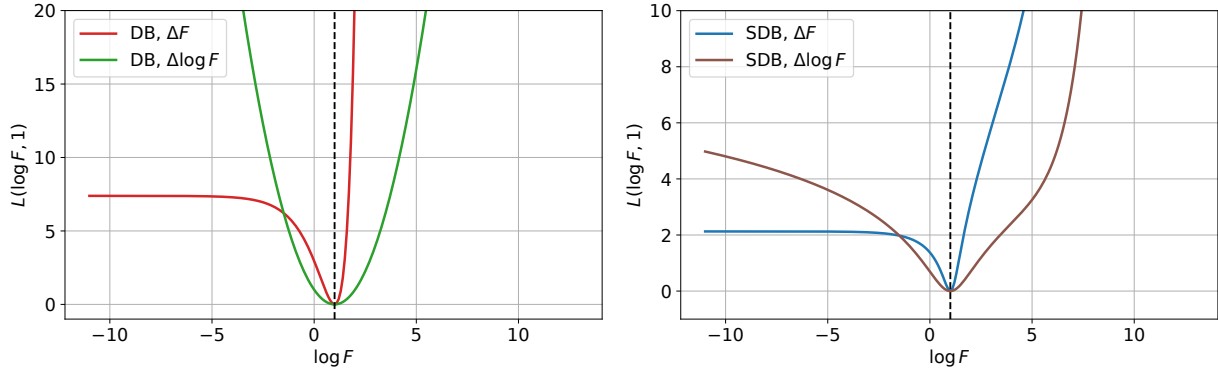

*Figure 3.* Plots for DB and SDB losses in $\Delta\mathcal{F}$ and $\Delta\log\mathcal{F}$ scales with fixed predicted log backward flow $= 1$ and varying predicted log forward flow. More specifically, green curve is $y = (x-1)^2$, red curve is $y = \left(e^x - e^1\right)^2$, brown curve is $y = \log\left(1 + (x-1)^2\right) \cdot (1 + 0.001e^x)$, blue curve is $y = \log\left(1 + \left(e^x - e^1\right)^2\right) \cdot (1 + 0.001e^x)$

.

The first is the difference between the predicted logarithms of the flows in the forward and backward direction $\log\mathcal{F}_F - \log\mathcal{F}_B$, while the second is the difference between predicted flows in the forward and backward direction $\mathcal{F}_F - \mathcal{F}_B$. Then, the standard DB loss (4) is

$$\mathcal{L}_{\text{DB}}(s \to s') = \Delta_{\log\mathcal{F}}(s, s', \theta)^2,$$

and the SDB loss (4) proposed in (Brunswic et al., 2024) is

$$\mathcal{L}_{\text{SDB}}(s \to s') = \log\left(1 + \varepsilon\Delta_{\mathcal{F}}(s, s', \theta)^2\right) \cdot (1 + \eta\mathcal{F}_\theta(s)).$$

However, for both losses, one can either replace $\Delta_{\log\mathcal{F}}$ with $\Delta_{\mathcal{F}}$ or the other way around. For visualization, let us fix the predicted log backward flow $\mathcal{F}_B$ to be, e.g., 1, and plot the losses with respect to the varying value of the predicted log forward flow $\mathcal{F}_F$. The plots are presented in Figure 3. One can note that as argument $\log\mathcal{F}_F$ decreases, both losses in $\Delta_{\mathcal{F}}$ scale quickly plateau, thus their derivative goes to zero. From the optimization perspective, this means that when the predicted log flow needs to be *increased*, the gradient step will be very small since the derivative of the loss is almost zero. On the other hand, when the predicted log flow needs to be *decreased*, the gradient step will be larger since losses have much higher derivatives in the corresponding regions. In combination with Proposition 3.12, this gives a possible explanation to the stability of $\Delta_{\mathcal{F}}$ scale losses: *they are biased towards underestimation of the flows and, as a result, biased towards solutions with smaller expected trajectory length*. We note that the same reasoning can be applied to the stable flow matching loss proposed in (Brunswic et al., 2024) since it also operates with differences between flows in $\Delta_{\mathcal{F}}$ scale.

However, as we show in our experimental evaluation (Section 4), *this comes at the cost of learning GFlowNets that match the reward distribution less accurately*.

### B.3. Fixed $\mathcal{P}_{\text{B}}$ and Trainable $\mathcal{P}_{\text{B}}$

In non-acyclic environments, $s_0$ and $s_f$ generally are fictive states that do not correspond to any object. Then $\mathcal{P}_{\text{F}}(s_f \mid s)$ corresponds to the probability to terminate a trajectory in state $s$, while $\mathcal{P}_{\text{F}}(s \mid s_0)$ corresponds to the probability that a trajectory starts in the state $s$. Thus, the choice of $\text{out}(s_0)$ is crucial in the design of the environment. If this set is large, e.g., coincides with $\in \mathcal{S} \setminus \{s_0, s_f\}$, one has to fix $\mathcal{P}_{\text{F}}(s \mid s_0)$ to some distribution, e.g. uniform, otherwise learning becomes intractable. However, in this case $\mathcal{P}_{\text{B}}(s_0 \mid s)$ has to be trainable, otherwise, it may be impossible to satisfy the detailed balance conditions for transitions $s_0 \to s$.

In our experiments, we consider two settings: training with a fixed $\mathcal{P}_{\text{B}}$ and using a trainable $\mathcal{P}_{\text{B}}$.

In case of fixed $\mathcal{P}_{\text{B}}$, we consider the case when $\text{out}(s_0) = \{s_{\text{init}}\}$, where $s_{\text{init}}$ is some fixed state $\in \mathcal{S} \setminus \{s_0, s_f\}$. Thus the first transition for all trajectories is to go from $s_0$ to $s_{\text{init}}$. Then, for any $s \in \mathcal{S} \setminus \{s_0, s_f, s_{\text{init}}\}$, $\mathcal{P}_{\text{B}}(\cdot \mid s)$ is uniform over the parents of $s$, while $\mathcal{P}_{\text{B}}(s_0 \mid s_{\text{init}}) = 1 - \varepsilon$ for some small $\varepsilon > 0$ and $\mathcal{P}_{\text{B}}(s \mid s_{\text{init}}) = \varepsilon/(\text{in}(s_{\text{init}}) - 1)$ for other transitions $s \to s_{\text{init}}$.

For a trainable $\mathcal{P}_\mathrm{B}$, we consider the case when $\mathrm{out}(s_0) = \mathcal{S} \setminus \{s_0, s_f\}$. Here we fix the first forward transition probability $\mathcal{P}_\mathrm{F}(s \mid s_0)$ to be uniform over $\mathcal{S} \setminus \{s_0, s_f\}$. In this case, DB loss for the first transition takes a special form:

$$\mathcal{L}_{\mathrm{DB}}(s_0 \to s) \triangleq \left( \log \mathcal{Z}_\theta - \log |\mathcal{S} \setminus \{s_0, s_f\}| - \log \mathcal{P}_\mathrm{B}(s_0 \mid s, \theta) - \log \mathcal{F}_\theta(s) \right)^2, \tag{16}$$

where $\log \mathcal{Z}_\theta - \log |\mathcal{S} \setminus \{s_0, s_f\}|$ corresponds to $\log \mathcal{F}_\theta(s_0) + \log \mathcal{P}_\mathrm{F}(s \mid s_0)$. An important note is that $\log \mathcal{F}_\theta(s_0)$ for optimal solutions always coincides with $\log \mathcal{Z}$; thus, it is usually harmful to apply state flow regularization (12) to it.

### B.4. Solving Small Environments Exactly

Suppose we have a fixed backward policy $\mathcal{P}_\mathrm{B}$ and a final flow $\mathcal{F}(s_f)$. Then, induced flows $\mathcal{F}$ and the corresponding forward policy $\mathcal{P}_\mathrm{F}$ can be obtained exactly for small environments. Consider the following system of linear equations with respect to $\hat{\mathcal{F}}(s)$ that arises from Proposition 3.6:

$$\begin{cases} \hat{\mathcal{F}}(s) = \sum_{s' \in \mathrm{out}(s)} \mathcal{P}_\mathrm{B}(s \mid s')\hat{\mathcal{F}}(s'), \ \forall s \in \mathcal{S} \setminus \{s_f\}, \\ \hat{\mathcal{F}}(s_f) = \mathcal{F}(s_f). \end{cases} \tag{17}$$

The system has $|\mathcal{S}|$ variables and $|\mathcal{S}|$ equations. $\hat{\mathcal{F}}(s) = \mathcal{F}(s)$ is a solution, where $\mathcal{F}(s)$ are state flows induced by $\mathcal{P}_\mathrm{B}$ and $\mathcal{F}(s_f)$, and the uniqueness of the solution follows from Proposition 3.7. Thus, by solving the system, one can exactly find induced state flows. Then, by Proposition 3.6 and Proposition 3.8, edge flows and $\mathcal{P}_\mathrm{F}$ can also be exactly expressed as

$$\mathcal{F}(s \to s') = \mathcal{P}_\mathrm{B}(s \mid s')\mathcal{F}(s'), \quad \mathcal{P}_\mathrm{F}(s' \mid s) = \mathcal{P}_\mathrm{B}(s \mid s')\mathcal{F}(s')/\mathcal{F}(s).$$

Finally, by Corollary 3.12, one can find the expected trajectory length of the induced trajectory distribution $\mathcal{P}$ as:

$$\mathbb{E}_{\tau \sim \mathcal{P}}[n_\tau] = \frac{1}{\mathcal{F}(s_f)} \sum_{s \in \mathcal{S} \setminus \{s_0, s_f\}} \mathcal{F}(s).$$

Interestingly, the system (17) can also be explained from the Markov Chain perspective. Let us take the graph $\mathcal{G}$ with reversed edges, add a loop from $s_0$ to itself, and use $\mathcal{P}_\mathrm{B}$ to define a Markov Chain with the following transition matrix: $P(s_0 \mid s_0) = 1$, $P(s \mid s') = \mathcal{P}_\mathrm{B}(s \mid s')$ if there is an edge $s \to s'$, and $P(s \mid s') = 0$ otherwise. It will be an absorbing Markov Chain, with an only absorbing state $s_0$ since it is reachable from any other state by Assumption 3.1. The transition matrix can be written in the following way:

$$P = \left[ \begin{array}{c|c} Q & R \\ \hline \mathbf{0} & 1 \end{array} \right],$$

where $Q$ is a $|\mathcal{S}| - 1$ by $|\mathcal{S}| - 1$ matrix and $R$ is a $|\mathcal{S}| - 1$ by $1$ matrix. Its fundamental matrix $N$, i.e., such a matrix that $N_{s,s'}$ is equal to the expected number of visits to a non-absorbing state $s'$ before being absorbed when starting from a non-absorbing state $s$, can be obtained as:

$$N = \sum_{k=0}^{+\infty} Q^k = (I - Q)^{-1},$$

where $I - Q$ is always invertible (Kemeny & Snell, 1969, Theorem 3.2.1). One can note that normalized flows $\mathcal{F}(s)/\mathcal{F}(s_f)$ coincide with the expected number of visits to $s$ when starting from $s_f$, thus coincide with the row of matrix $N$ corresponding to $s_f$. Finally, notice that $(I - Q)$ coincides with the transposed matrix of the truncated system (17) (with the exception of the variable and the equation corresponding to $s_0$), thus, such system has a unique solution $\mathcal{F}(s_f)(I - Q)^{-T} e_{s_f} = \mathcal{F}(s_f)N^T e_{s_f}$, where $e_{s_f}$ is a vector of size $|\mathcal{S}| - 1$ that has $1$ on the position corresponding to $s_f$ and $0$ on all others. The variable corresponding to $s_0$ should be handled separately, but it is easy to see $\hat{\mathcal{F}}(s_0) = \sum_{s' \in \mathrm{out}(s)} \mathcal{P}_\mathrm{B}(s \mid s')\mathcal{F}(s') = \mathcal{F}(s_0)$.

## C. Experimental Details

### C.1. Loss Choice

While (Brunswic et al., 2024) used the original flow matching loss (Bengio et al., 2021) for experimental evaluation, it was previously shown to be less computationally efficient and provide slower convergence than other GFlowNet losses (Malkin

et al., 2023; Madan et al., 2023) in the acyclic case, so we carry out experimental evaluation with the more broadly employed detailed balance loss (Bengio et al., 2023). Moreover, flow matching loss does not admit explicit parameterization of a backward policy, as well as training with fixed backward policies, thus not allowing us to study some of the phenomena we explore in the experiments.

In addition, we note that the proposed state flow regularization (12) can be potentially applied with other GFlowNet losses that learn flows, e.g. `SubTB` (Madan et al., 2023), or with the modification of `DB` proposed in (Deleu et al., 2022) that implicitly parametrizes flows as $\mathcal{F}(s) = \mathcal{R}(s)/\mathcal{P}_\mathrm{F}(s_f \mid s)$.

### C.2. Hypergrids

Formally, $\mathcal{S} \setminus \{s_0, s_f\}$ is a set of points with integer coordinates inside a $D$-dimensional hypercube with side length $H$: $\left\{\left(s^1, \ldots, s^D\right) \mid s^i \in \{0, 1, \ldots, H-1\}\right\}$. $s_0$ and $s_f$ are auxiliary states that do not correspond to any point inside the grid. Possible transitions correspond to increasing or decreasing any coordinate by 1 without exiting the grid. In addition, for each state $s \in \mathcal{S} \setminus \{s_0, s_f\}$ there is a terminating transition $s \to s_f$. GFlowNet reward at $s = (s^1, \ldots, s^D)$ is defined as

$$\mathcal{R}(s) \triangleq R_0 + R_1 \prod_{i=1}^{D} \mathbb{I}\left[0.25 < \left|\frac{s^i}{H-1} - 0.5\right|\right] + R_2 \prod_{i=1}^{D} \mathbb{I}\left[0.3 < \left|\frac{s^i}{H-1} - 0.5\right| < 0.4\right],$$

where $0 < R_0 \ll R_1 < R_2$. (Brunswic et al., 2024) do not specify reward parameters used in their experiments, so we use the parameters from the acyclic version of the environment studied in (Malkin et al., 2022), i.e. $(R_0 = 10^{-3}, R_1 = 0.5, R_2 = 2.0)$.

The utilized metric is:

$$\frac{1}{2} \sum_{x \in \mathcal{X}} |\mathcal{R}(x)/\mathcal{Z} - \pi(x)|,$$

where $\pi(x)$ is the empirical distribution of last $2 \cdot 10^5$ samples seen in training (endpoints of trajectories sampled from $\mathcal{P}_\mathrm{F}$).

All models are parameterized by MLP with 2 hidden layers and 256 hidden size, which accept a one-hot encoding of $s$ as input. $\mathcal{F}_\theta(s), \mathcal{P}_\mathrm{F}(s' \mid s, \theta), \mathcal{P}_\mathrm{B}(s \mid s', \theta)$ share the same backbone, with different linear heads predicting the logarithm of the state flow, the logits of the forward policy and the logits of the backward policy. In the case of the fixed $\mathcal{P}_\mathrm{B}$, $s_\mathrm{init}$ corresponds to the center of the grid, and we take $\varepsilon = 10^{-8}$ (see Appendix B.3).

We train all models on-policy. We use Adam optimizer with a learning rate of $10^{-3}$ and a batch size of 16 (number of trajectories sampled at each training step). For $\log \mathcal{Z}_\theta$ we use a larger learning rate of $10^{-2}$ (see (Malkin et al., 2022)). All models are trained until $2 \cdot 10^6$ trajectories are sampled, and the empirical sample distribution $\pi(x)$ is computed over the last $2 \cdot 10^5$ samples seen in training. For `SDB` we set $\varepsilon = 1.0$ and $\eta = 10^{-3}$. We found that using larger values of $\eta$ can lead to smaller expected trajectory length, but also significantly interfere with the sampling fidelity of the learned GFlowNet, thus we opt for these values in our experiments.

### C.3. Permutations

All models are parameterized by MLP with 2 hidden layers and 128 hidden size, which accept a one-hot encoding of $s$ as input. $\mathcal{F}_\theta(s), \mathcal{P}_\mathrm{F}(s' \mid s, \theta), \mathcal{P}_\mathrm{B}(s \mid s', \theta)$ share the same backbone, with different linear heads predicting the logarithm of the state flow, the logits of the forward policy and the logits of the backward policy. In the case of the fixed $\mathcal{P}_\mathrm{B}$, $s_\mathrm{init}$ corresponds to the permutation $(n, n-1, \ldots, 2, 1)$, and we take $\varepsilon = 10^{-8}$ (see Appendix B.3).

We train all models on-policy. We use Adam optimizer with a learning rate of $10^{-3}$ and a batch size of 512 (number of trajectories sampled at each training step). We found that using small batch sizes can significantly hinder training stability for this environment; thus, we opt for a larger value. All models are trained for $10^5$ iterations. For $\log \mathcal{Z}_\theta$ we use a larger learning rate of $10^{-2}$ (see (Malkin et al., 2022)). To compute $\Delta \log \mathcal{Z}$ we take the average value of $\log \mathcal{Z}_\theta$ over the last 10 training checkpoints. Empirical distribution $\hat{C}(k)$ is computed over the last $10^5$ samples seen in training. For `SDB` we set $\varepsilon = 1.0$ and $\eta = 10^{-3}$. We found that using larger values of $\eta$ can lead to smaller expected trajectory length, but also significantly interfere with the sampling fidelity of the learned GFlowNet, thus we opt for these values in our experiments.

Suppose that $x_1, \ldots, x_m$ is a set of GFlowNet samples (terminal states of trajectories sampled from $\mathcal{P}_\mathrm{F}$). Then, the empirical

$L_1$ error of fixed point probabilities is defined as:

$$\sum_{k=0}^{N}\left|C(k) - \frac{1}{m}\sum_{i=1}^{m}\mathbb{I}\{x_i(k) = k\}\right|,$$

and the relative error of mean reward is defined as

$$\left|\frac{\mathbb{E}[\mathcal{R}(x)] - \frac{1}{m}\sum_{i=1}^{m}\mathcal{R}(x_i)}{\mathbb{E}[\mathcal{R}(x)]}\right|,$$

where $\mathbb{E}[\mathcal{R}(x)] = \sum_{x\in\mathcal{X}}\mathcal{R}(x)\frac{\mathcal{R}(x)}{\mathcal{Z}}$. We compute mean reward over $10^4$ samples.

### C.3.1. REWARD DISTRIBUTION PROPERTIES

We define the GFlowNet reward as $\mathcal{R}(s) = \exp(\frac{1}{2}\sum_{k=1}^{n}\mathbb{I}\{s(k) = k\})$. We are interested in the true values of three quantities:

1. normalizing constant $\mathcal{Z} = \sum_{x\in\mathcal{X}}\mathcal{R}(x)$,

2. true expected reward $\mathbb{E}[\mathcal{R}(x)] = \sum_{x\in\mathcal{X}}\mathcal{R}(x)\frac{\mathcal{R}(x)}{\mathcal{Z}}$,

3. fixed point probabilities $C(k) = \mathbb{P}((\sum_{i=1}^{n}\mathbb{I}\{x(i) = i\}) = k)$ with respect to the reward distribution.

While computing sums over all permutations is intractable for $n$ above some threshold, below, we show that for this particular reward, analytical expressions for these quantities can be derived.

First, we will derive the formula for the total number of permutations of length $n$ with exactly $k$ fixed points, which we will denote as $D(k,n)$. In combinatorics, such permutations are known as partial derangements, and the quantity is known as rencontres numbers (Comtet, 1974, p.180). Note that

$$D(k,n) = \binom{n}{k}D(0, n-k),$$

since choosing a permutation with $k$ fixed points coincides with choosing $k$ positions for fixed points, and permuting the remaining elements such that there are no fixed points among them. So let us start with the derivation of $D(0,n)$. Denote $S_i$ to be the set of permutations on $n$ elements that has a fixed point on position $i$. Then, by the inclusion-exclusion principle, we have

$$|S_1 \cup \cdots \cup S_n| = \sum_{i}|S_i| - \sum_{i<j}|S_i \cap S_j| + \sum_{i<j<k}|S_i \cap S_j \cap S_k| + \cdots + (-1)^{n+1}|S_1 \cap \cdots \cap S_n|$$

$$= \binom{n}{1}(n-1)! - \binom{n}{2}(n-2)! + \binom{n}{3}(n-3)! - \cdots + (-1)^{n+1}\binom{n}{n}0!$$

$$= \sum_{i=1}^{n}(-1)^{i+1}\binom{n}{i}(n-i)! = n!\sum_{i=1}^{n}\frac{(-1)^{i+1}}{i!}.$$

Then

$$D(0,n) = n! - |S_1 \cup \cdots \cup S_n| = n! - n!\sum_{i=1}^{n}\frac{(-1)^{i+1}}{i!} = n!\sum_{i=0}^{n}\frac{(-1)^{i}}{i!}.$$

Thus, we have

$$D(k,n) = \binom{n}{k}D(0, n-k) = \frac{n!}{k!(n-k)!}(n-k)!\sum_{i=0}^{n-k}\frac{(-1)^{i}}{i!} = n!\sum_{i=0}^{n-k}\frac{(-1)^{i}}{i!k!}.$$

Finally, all of the quantities we are interested in are easily expressed through $D(k,n)$:

1. $\mathcal{Z} = \sum_{k=0}^{n}D(k,n)\exp(k/2)$.

2. $\mathbb{E}[\mathcal{R}(x)] = \sum_{k=0}^{n}D(k,n)\exp(k/2)\frac{\exp(k/2)}{\mathcal{Z}}$.

3. $C(k) = D(k,n)\frac{\exp(k/2)}{\mathcal{Z}}$.

For reference, the formula yields values of $\log\mathcal{Z} \approx 3.8262, 11.2533, 42.9843$ for $n = 4$, $n = 8$, and $n = 20$ respectively.

# D. Additional Plots

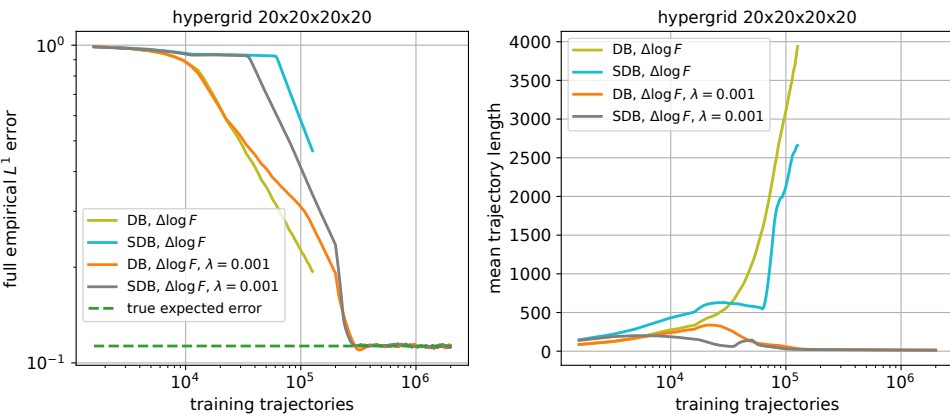

*Figure 4. Left:* evolution of $L^1$ distance between empirical distribution of samples and target distribution. *Right:* evolution of mean length of sampled trajectories. Here we note that when $\Delta \log \mathcal{F}$ scale losses are employed without state flow regularization, mean trajectory length tends to infinity. Plots are not full since training is done on-policy. Thus, the time needed for full training also grows according to the length of trajectories.

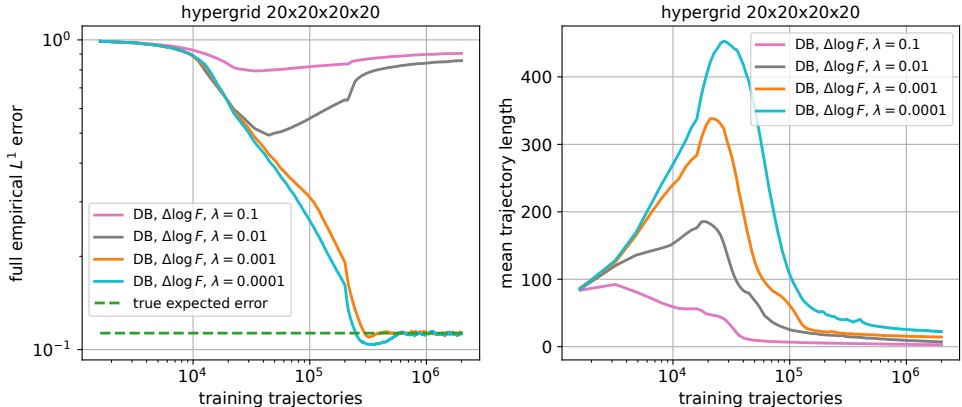

*Figure 5. Left:* evolution of $L^1$ distance between empirical distribution of samples and target distribution. *Right:* evolution of mean length of sampled trajectories. Here, we see the effects of state flow regularization of different strength $\lambda$. Larger values of $\lambda$ lead to smaller mean trajectory length, however, if $\lambda$ is too large, the obtained forward policy will be significantly biased.

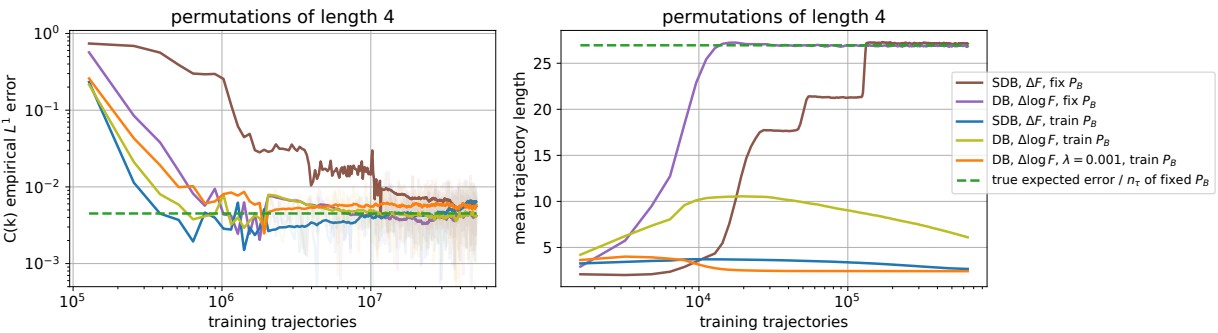

*Figure 6.* Comparison of non-acyclic GFlowNet training losses on a small permutation environment. *Left:* evolution of $L_1$ distance between true and empirical distribution of fixed point probabilities $C(k)$. *Right:* evolution of mean length of sampled trajectories. The results are similar to the same experiment on hypergrids (Figure 1), with the only difference that here SDB loss in $\Delta \mathcal{F}$ scale here has fast convergence with a trainable backward policy.

