# OpenReview forum: "Revisiting Non-Acyclic GFlowNets in Discrete Environments"
_ICML.cc/2025/Conference — ICML 2025 poster_

### Official Review · Reviewer_6mHF · 2025-03-14

**Overall Recommendation:** 4

**Summary:**

The paper introduces a new theory of Non-Acyclic GFlowNet. The main intuition is that in the non-acyclic case, the visiting probability of an edge doesn't follow the flow matching constraint, because it doesn't take into account the cycles. Rather, the expected number of visits satisfies the FM, making it the appropriate quantity to define a flow. Another important idea is that the length of the trajectories is proportional to the total states flow.

**Claims And Evidence:**

All the claims are mathematically proven in appendix. Experimental evidence supports the provided theory.

**Essential References Not Discussed:**

not to my knowledge

**Experimental Designs Or Analyses:**

all good

**Methods And Evaluation Criteria:**

The paper evaluates its methodology on two distinct tasks: non-acyclic hypergrid and permutations. Hypergrid is widely recognized and frequently used in Generative Flow Network research. The permutation task was first presented by Brunswic et al. (2024), though this paper implements it with a simplified reward structure. Both the selected tasks and the evaluation metrics are appropriate and meaningful for assessing the method's performance.

**Other Comments Or Suggestions:**

NA

**Other Strengths And Weaknesses:**

Strength: The paper is very well written!

Weaknesses: The paper could benefit from more experimental settings

**Questions For Authors:**

- Can this theoretical framework be extended to continuous state spaces without significant modifications, or would it require fundamental adaptations?

- If we constrain training trajectories to be acyclic (either by preventing cycles during sampling or by truncating cyclic segments post-sampling), does this approach improve training stability and convergence? Furthermore, does such training naturally lead to fewer cycles during inference, even when regularization mechanisms are removed?

**Relation To Broader Scientific Literature:**

Brunswic et al. (2024) were the first who addressed non-acyclic GFlowNets, introducing the concept of stable loss and developing stabilized versions of conventional training objectives. The current paper refines several assertions made by Brunswic et al., specifically: clarifying that visitation probability adheres to flow matching, writing Proposition 3.11 to show it represents an equality rather than merely an inequality, and demonstrating that non-acyclic GFlowNets can be effectively trained using unstable losses when proper regularization is applied. Additionally, this work expands upon Tiapkin et al. (2024) by establishing connections between GFlowNets and Entropy-Regularized Reinforcement Learning within non-acyclic contexts.

**Theoretical Claims:**

all good

---

> ### Author Rebuttal · Authors · 2025-03-31
>
> We highly appreciate the feedback from the reviewer and are happy to answer their questions. We are pleased that the reviewer acknowledged the contributions of our paper and found it very well written.
>
> Firstly, we would like to note that we actually implement the permutation task with a more complex reward function compared to [1], not with a simpler one (see lines 366-374 in Section 4.3 and Appendix C.3.1 for further details).
>
> Secondly, the case of continuous state spaces was explored in [1], while the goal of our work was to focus on the discrete case and show how the theory and results of [1] can be simplified and expanded in discrete environments. While our own construction can be potentially extended to the continuous case following an approach similar to the one introduced in [2], it will indeed require fundamental adaptations, i.e., considering general measurable spaces instead of finite graphs. An interesting question here is whether the refined results we demonstrate in our paper compared to [1] (e.g. nature of flows, training with fixed backward policies, equality instead of inequality in Proposition 3.11) can be demonstrated in the continuous case as well, or whether they are specific to the discrete case and do not hold/need a different interpretation  for continuous spaces. We consider this to be a crucial direction for future research.
>
> Finally, we agree that the idea to truncate/prevent cycles in trajectories used for training seems natural. In addition, we note that [1] followed a similar approach, but bounded the maximum length of trajectories used in training instead of removing cycles from them. However, this may come with a number of potential issues. Using a distribution over trajectories in training that is different from the one parameterized by the forward policy will mean that the training is done off-policy, which is a viable choice for GFlowNet algorithms, and it is indeed possible that truncating cycles during training will lead to improved stability. Yet, during inference, if one wants to obtain samples from the target distribution using the learned policy, one has to faithfully run the generation process following the policy until a terminating transition is sampled, and using some form of truncation may lead to a bias. In addition, even if the truncation is used in training, there is no guarantee how the true expected trajectory length of the learned policy will behave if the regularization is removed. We hypothesize that it will still be growing if a $\Delta \log F$ scale loss is used without regularization (like in Figure 4), but it is difficult to say for sure without explicit empirical validation of different truncation strategies. We also find this to be an interesting direction for future research.
>
> References:\
> [1] Brunswic et al. A Theory of Non-Acyclic Generative Flow Networks. AAAI 2024\
> [2] Lahlou et al. A theory of continuous generative flow networks. ICML 2023

---

### Official Review · Reviewer_c5qu · 2025-03-15

**Overall Recommendation:** 4

**Summary:**

This paper offers a new perspective on the theory of GFlowNets in the cas where the state space is no longer acyclic. The author analyzed an existing theory of non-acyclic GFlowNets and provide insights as to when their proposed approach is necessary, and show that under an appropriate regularization, all existing losses used in the GFlowNet literature transfer (which were exclusively derived for acyclic state spaces) to the non-acyclic case.

**Claims And Evidence:**

Claims are supported by clear and convincing evidence.

**Essential References Not Discussed:**

Not an essential reference, especially since this was released on Arxiv only a few weeks before the submission deadline, but the constrained optimization formulation in Eq 11 (motivating the flow regularization) is similar to Proposition 4.1.3 of (Deleu, 2025). This result appears in a 200 pages PhD thesis and does not seem to have been published prior to this. The authors should only include that if they deem necessary.

---

Deleu, T. Generative Flow Networks: Theory and Applications to Structure Learning. 2025.

**Experimental Designs Or Analyses:**

The experimental designs and analyses are sound.

**Methods And Evaluation Criteria:**

The proposed method makes sense for the problem at hand.

**Other Comments Or Suggestions:**

- Lines 101-102: This is not what the reward matching condition corresponds to. The reward matching condition of [Bengio et al., 2023] corresponds to $F(x \rightarrow s_{f}) = R(x)$ (and its variant $F(x)P_{F}(s_{f}\mid x) = R(x)$ in the case of the DB condition). In fact, this is exactly the condition later stated in line 74 or 86, and (almost) correctly stated in eq 9. The condition in the submission $P(\{\tau \mid x\in\tau\}) = R(x) / Z$ is the desiderata of GFlowNets (i.e., its terminating state distribution is proportional to $R$). Note that even $P(\{\tau \mid x\in\tau\})$ is *not* the terminating state distribution: the terminating state distribution is the marginal over trajectories that terminate in $x$, in the sense mentioned in line 89.
- Typo $s$ instead of $s’$ in line 125, $s’$ instead of $s$ in line 128.
- Typo $R(s)$ instead of $R(s_f)$ in eq 9 and line 249.

---

Bengio, Y., Lahlou, S., Deleu, T., Hu, E. J., Tiwari, M., and Bengio, E. GFlowNet Foundations. JMLR, 2023

**Other Strengths And Weaknesses:**

N/A

**Questions For Authors:**

- It seems like in many applications involving GFlowNets, the state space is a design choice made by the practitioners which as full control over, as long as the terminal states are the elements of the sample space. This is very different from RL, where the MDP is often modeling the real-world that we don't have much control over. Therefore, it seems like in most cases the state space can be designed to be acyclic, where the standard theory of GFlowNets hold and no considerations regarding the expected length of the trajectories is necessary. Case in point, the two environments studied in this submission (which are based on prior work) are designed specifically to be non-acyclic; but we could very well design them being acyclic (hypergrid is often studied in the acyclic case). Can you give a practical example of an environment where non-acyclicity is necessary for GFlowNets?

**Relation To Broader Scientific Literature:**

This submission challenges some findings made by (Brunswic et al., 2024), and provides a new perspective on GFlowNets in the case where the state space is non-acyclic anymore. They show that in the case where $P_{B}$ is fixed, the stable versions of the losses proposed by (Brunswic et al., 2024) is not necessary. They also show a stronger result regarding the expected length of the trajectories (in Proposition 3.11, as opposed to an inequality in prior work). They state and empirically verify a "scaling hypothesis" that shows that prior losses working in flow space as in (Brunswic et al., 2024) lead to lower expected trajectory lengths, but often fail to get a good approximation of the target distribution.

---

Brunswic, L., Li, Y., Xu, Y., Feng, Y., Jui, S., and Ma, L. A theory of non-acyclic generative flow networks. AAAI 2024.

**Theoretical Claims:**

The theoretical claims are correct.

---

> ### Author Rebuttal · Authors · 2025-03-31
>
> We are grateful to the reviewer for their constructive feedback and valuable suggestions, and are happy to provide further details.
>
> Firstly, we thank the reviewer for the provided reference to [1]. While Proposition 4.1.3 does not state minimality of the expected trajectory length of the solution, it does indeed present a similar constrained optimization problem to the one introduced in Eq. 11 in our work.
>
> Secondly, we agree with the comments and suggestions regarding the formulation of the reward matching condition, and thank the reviewer for pointing out typos and inconsistencies related to it. We note that depending on the adopted structure of the environment in acyclic GFlowNet construction, reward matching can be formulated differently. If terminal states in the graph have no outgoing edges and their set coincides with the target space of interest $X$, the reward matching condition is formulated as $F(x) = R(x)$, e.g., see [2]. However, if one considers the graph structure with a sink state $s_f$, which is the construction adopted in our work, and terminating transitions $x \to s_f$ indicate the end of the trajectory (rather than reaching a terminal state), the reward matching condition has to be formulated as $F(x \to s_f) = R(x)$ because the state $x$ can both happen to be in the middle or in the end of a trajectory. In this case, $P(\tau \mid x \in \tau)$ indeed is not the terminating state distribution, as pointed out by the reviewer. We will incorporate the suggested fixes in the revision of our paper.
>
> Finally, we agree that the applicability of non-acyclic GFlowNets is a crucial question. There are several examples of such applications mentioned in the introduction of our paper, but we recognize that putting more emphasis on them will further strengthen the presentation and motivation of our work, and we will gladly incorporate this in the paper. Below we provide a more detailed discussion of some examples:
>
> 1. One example, which was also discussed in [3], is related to modeling distributions over objects with intrinsic symmetries. Consider a class of environments where states are elements of some group, and transitions are given via a generating set of this group, thus corresponding to applying the group operation on the current state and some element of the generating set. An example is an environment induced by Rubik’s cube, where states are possible arrangements of a Rubik’s cube, and actions correspond to rotating a face of a cube. Another one is the permutation task studied in our paper (Section 4.3). While in some cases an acyclic environment can be designed to generate group elements, such environments of “algebraic” origin naturally contain cycles, thus falling under the area of our study.
> 2. Next example is related to potential use cases of GFlowNets where one is more interested not in the objects sampled from the reward distribution, but in the trained policy itself. One can consider RL environments where the reward is sparse and given only at the end of a trajectory, but similarly to GFlowNets formulate a problem of finding a policy that will end up in terminal states with probabilities proportional to the exponent of the reward, thus promoting diversity and exploration. Most RL environments contain cycles, and the GFlowNet environments considered in our work are almost arbitrary finite discrete deterministic MDPs with sparse rewards. Thus, our work makes another step towards bridging two research fields. Moreover, the approach presented in our work aims at finding a policy with the smallest expected trajectory length, which may be of additional interest in RL applications.
> 3. Finally, one can consider regular acyclic environments where GFlowNets are utilized, e.g. molecular graph generation environments, but introduce an option to remove some part of the current object during generation, not only add new parts, leading to cycles in the environment. This idea is also briefly mentioned in [1]. A possible intuition behind this design choice is allowing the model to correct “mistakes” it can potentially make in the generation process, thus leading to a more expressive class of policies.
>
> References:\
> [1] Tristan Deleu. Generative Flow Networks: Theory and Applications to Structure Learning. 2025\
> [2] Malkin et al. Trajectory balance: Improved credit assignment in GFlowNets. NeurIPS 2022\
> [3] Brunswic et al. A Theory of Non-Acyclic Generative Flow Networks. AAAI 2024

---

### Official Review · Reviewer_Mo9C · 2025-03-17

**Overall Recommendation:** 4

**Summary:**

This paper revisits the theoretical framework of Generative Flow Networks (GFlowNets) in non-acyclic discrete environments, where the traditional assumption of acyclicity is relaxed. The authors propose a simplified and more intuitive formulation of non-acyclic GFlowNets that extends prior work by Brunswic et al. (2024). The contributions of the paper include:

- A more straightforward theory for non-acyclic GFlowNets, avoiding more complex measure-theoretical framing for discrete state-space contexts. Furthermore, the theory explores the connection between Entropy-regularized RL and GFlowNets, extending previously known results to the non-acyclic setting.
- Refining the understanding of loss stability and convergence:
  - Challenging the need for modified stable losses when the backward policy (PB) is fixed.
  - It proves that when PB is trainable, minimizing the expected trajectory length is equivalent to minimizing the total flow.
  - Introducing state flow regularization as a means to stabilize non-acyclic GFlowNet training.
- Meaningful empirical results validate the theory and explore an interesting question regarding the scaling effects of loss functions and trajectory length control (log-scale errors impact expected trajectory length).

Overall, the paper combines key novel theoretical results and simplifies previous results with a combination of theoretical and empirically grounded work.

**Claims And Evidence:**

| **Claim** | **Evidence** |
| --- | --- |
| Non-acyclic GFlowNets can be formulated with a simpler and more intuitive framework than prior work. | The authors introduce a discrete-state formulation, avoiding measure-theoretic complexity. |
| A fixed backward policy removes the need for loss stability conditions. | Theoretical results and empirical evaluations show that standard training remains stable under a fixed PB. |
| When PB is trainable, minimizing expected trajectory length is equivalent to minimizing total flow. | The authors prove this equivalence mathematically. |
| ∆F and ∆logF losses effects on trajectory length | Experiments show that ∆F and ∆logF losses estimated trajectory length and metrics. |
| GFlowNets in non-acyclic environments are equivalent to entropy-regularized RL. | A generalization of Tiapkin et al. (2024) proves this connection formally. |

**Essential References Not Discussed:**

The paper provides a strong theoretical foundation for non-acyclic GFlowNets. Still, it could further contextualize its contributions within recent advances in backward training, generalized training strategies, and correctness of learned distributions.

1. Backward Training in GFlowNets
Given that the paper investigates the role of PB in non-acyclic formulations, discussing these approaches would strengthen its connection to existing literature:
  - *Pessimistic Backward Policy for GFlowNets* (Jang et al., NeurIPS 2024) i
  - *Looking Backward: Retrospective Backward Synthesis for Goal-Conditioned GFlowNets* (He et al.)
  - *Optimizing Backward Policies via Trajectory Likelihood Maximization* (Gritsaev et al.)

2. Generalized Training Strategies and Divergence Losses
Beyond standard log-squared losses, recent works explore alternative loss functions for GFlowNets:
  - *Beyond Squared Error: Exploring Loss Design for Enhanced Training of Generative Flow Networks* (Hu et al., ICLR 2025)
  - *On Divergence Measures for Training GFlowNets* (Silva et al., NeurIPS 2024)

3. Correctness of Learned Distributions
  - *When do GFlowNets learn the right distribution?* (Silva et al., ICLR 2025) examines the stability of GFlowNets under balance violations and introduces the Flow Conservation Score (FCS) metric.

**Experimental Designs Or Analyses:**

The paper presents well-structured experiments evaluating the impact of different loss functions, backward policies, and trajectory length control in non-acyclic GFlowNets. The chosen environments (hyper grid and permutation-based tasks) are standard benchmarks for testing GFlowNet behavior, and the experiments effectively highlight the role of loss stability, scaling effects, and state flow regularization.

Key strengths of the experimental design:
- Controlled comparisons between different loss formulations (∆F vs. ∆logF).
- Analysis of trajectory length effects, demonstrating how different regularization methods impact stability.
- Clear ablations on the role of the backward policy (fixed vs. trainable).

However, the paper lacks experiments on larger, real-world applications. Overall, the experimental design is well-motivated and supports the theoretical claims, but further empirical validation on complex, real-world tasks would make the findings more compelling.

**Methods And Evaluation Criteria:**

The proposed methods and evaluation criteria align well with the problem of training non-acyclic GFlowNets. The authors use well-established benchmark datasets, such as hypergrid and permutation-based environments, appropriate for validating their theoretical claims. The evaluation setup effectively tests the impact of different loss functions, backward policies, and flow stability in non-acyclic settings.

However, additional experiments on larger-scale, real-world applications (e.g., molecule generation, Bayesian structure learning) would further strengthen the paper’s empirical contributions.

**Other Comments Or Suggestions:**

NA

**Other Strengths And Weaknesses:**

In my view, the main weakness is that non-acyclic GFlowNets are not well motivated. The reader unfamiliar with GFlowNets literature might see this as simply theoretical details of the theory. Connecting these ideas to broader ideas in ML and probabilistic models, particularly issues regarding inference and representational power of graphical models with and without cycles (e.g. Bayesian Nets, Markov Random Fields; Loopy BP, Bethe Approximation) could give a better context to why care about cycles.

Overall, I find this paper interesting and insightful for generative modeling researchers, introducing simpler and more intuitive theory for non-acyclic GFlowNets.

**Questions For Authors:**

1. Impact of non-acyclic formulation on generalized divergence training
   - Recent works [1,2] have explored training using f-divergence objectives, including variance reduction techniques, expanding the connection with variational inference.
   - How does the non-acyclic formulation impact variance reduction techniques for divergence-based training?
   - Could state flow regularization interact with generalized divergence-based objectives, such as f-divergences, in an on-policy setting?

2. Effect of log-scale errors on generalized loss functions
   - The paper presents insights into loss scaling in non-acyclic GFlowNets, for example, showing that ∆logF losses require flow regularization.
   - Recent studies [2] suggest that different loss functions correspond to different divergence measures, affecting *exploration vs. exploitation trade-offs*.
   - Could the results on *log-scale loss* guide designing *alternative divergence-based losses* beyond squared error?

3. Correct distributional learning:
   - Recently, the stability of GFlowNet to minor flow errors has been studied, and a flow consistency in sub-graphs (FCS) [3] metric has been proposed to evaluate whether GFlowNets genuinely learn the correct distribution.
   - Does the non-acyclic formulation lead to systematic balance violations in some settings?
   - How could the FCS metric (or similar ideas of distributional consistency for subgraphs) be applied to evaluate non-acyclic GFlowNets?

4. From a practical point of view, why would someone choose to use a non-acyclic formulation of GFlowNet rather than the acyclic one, given that a large enough acyclic state graph could represent most distributions of interest?

These questions aim to clarify how the proposed non-acyclic formulation interacts with a wider variety of training strategies (generalized divergences, variance reduction), the theoretical aspects of distributional correctness in GFlowNets, and the practical implications of this formulation.

References:
[1] Silva et al. On Divergence Measures for Training GFlowNets. NeurIPS 2024.
[2] Hu et al.. Beyond Squared Error: Exploring Loss Design for Enhanced Training of Generative Flow Networks. ICLR 2025.
[3] Silva et al. When do GFlowNets learn the right distribution? ICLR 2025.

**Relation To Broader Scientific Literature:**

The paper simplifies and expands on two key previous theoretical results for GFlowNet, particularly the theory of non-acyclic GFlowNets and the connection with entropy-regularized RL. It expands ideas on the role of fixed and trained backward policy in the stability of GFlowNet training in this context and provides broad empirical evidence. It has the potential to clarify existing practices and provide theoretical grounding for algorithmic design.

**Theoretical Claims:**

Yes, the theoretical claims were checked, and the proofs appear correct.

The paper provides a well-structured derivation of non-acyclic GFlowNet, proving key results related to trajectory length minimization, state flow regularization, and equivalence to entropy-regularized RL. The mathematical arguments are logically consistent and align with prior GFlowNet research.

The results are a meaningful extension of Brunswic et al. (2024) and Tiapkin et al. (2024), and I did not find any critical flaws in the derivations.

---

> ### Author Rebuttal · Authors · 2025-03-31
>
> We want to express our gratitude to the reviewer for the very detailed review and valuable suggestions. We are pleased that the reviewer acknowledged the contributions of our paper, novelty and soundness of our theoretical results, as well as the strengths of our experimental design.
>
> Regarding the weakness outlined by the reviewer (and question 4), we agree that the motivation and applicability of non-acyclic GFlowNets are crucial points. In addition to possible ideas related to Bayesian networks and graphical models mentioned by the reviewer, below we present a number of potential use cases that seem highly relevant from our perspective. While some of them are mentioned in the introduction of our paper, we recognize that putting more emphasis on this will further strengthen the presentation and motivation of our work.
>
> 1. One example, which was also discussed in [1], is related to modeling distributions over objects with intrinsic symmetries. Consider a class of environments where states are elements of some group, and transitions are given via a generating set of this group, thus corresponding to applying the group operation on the current state and some element of the generating set. An example is an environment induced by Rubik’s cube, where states are possible arrangements of a Rubik’s cube, and actions correspond to rotating a face of a cube. Another one is the permutation task studied in our paper (Section 4.3). While in some cases an acyclic environment can be designed to generate group elements, such environments of “algebraic” origin naturally contain cycles, thus falling under the area of our study.
> 2. Next example is related to potential use cases of GFlowNets where one is more interested not in the objects sampled from the reward distribution, but in the trained policy itself. One can consider RL environments where the reward is sparse and given only at the end of a trajectory, but similarly to GFlowNets formulate a problem of finding a policy that will end up in terminal states with probabilities proportional to the exponent of the reward, thus promoting diversity and exploration. Most RL environments contain cycles, and the GFlowNet environments considered in our work are almost arbitrary finite discrete deterministic MDPs with sparse rewards. Thus, our work makes another step towards bridging two research fields. Moreover, the approach presented in our work aims at finding a policy with the smallest expected trajectory length, which may be of additional interest in RL applications.
> 3. One can consider regular acyclic environments where GFlowNets are utilized, e.g. molecular graph generation environments, but introduce an option to remove some part of the current object during generation, not only add new parts, leading to cycles in the environment. A possible intuition behind this design choice is allowing the model to correct “mistakes” it can potentially make in the generation process, thus leading to a more expressive class of policies.
>
> Next, while we do cite some of the references mentioned by the reviewer, we agree that including a more active discussion in the paper on connections to previous GFlowNet literature will help further contextualize our contributions. We thank the reviewer for the suggestion and the proposed references, and will incorporate them in a revision of our paper.
>
> Finally, we highly appreciate the detailed questions from the reviewer and are happy to answer them. Firstly, we note that the proposed state flow regularization could indeed be potentially applied with different divergence-based objectives. The applicability of the proposed regularization depends on the utilized parameterization of a GFlowNet rather than the exact divergence used to compute the loss. The parameterization has to include the state (or edge) flow, so our regularization cannot be applied with trajectory-level losses that only involve learning $P_F$ and $P_B$, but can be applied with transition and sub-trajectory level losses that also include learning of the flows. Secondly, stable losses in [1] were also derived from specific $f$-divergences, so the analysis of [2] could indeed be potentially extended to explain exploration vs exploitation trade-offs in non-acyclic GFlowNets, as well as provide a deeper theoretical analysis of our scaling hypothesis. Finally, we believe that there should be no issue in applying the FCS metric in the non-acyclic case, as Equation 8 from [3] still holds for our construction. In addition, extending the analysis of [3] on propagation of errors in flow networks to the non-acyclic case will be a non-trivial task, thus posing another interesting direction for future research.
>
> References:\
> [1] Brunswic et al. A Theory of Non-Acyclic Generative Flow Networks. AAAI 2024\
> [2] Hu et al. Beyond Squared Error: Exploring Loss Design for Enhanced Training of Generative Flow Networks. ICLR 2025\
> [3] Silva et al. When do GFlowNets learn the right distribution? ICLR 2025

---

### Decision · Program_Chairs · 2025-05-01

**Decision:**

Accept (poster)

**Comment:**

All the reviewers agree that the paper should be accepted, and so do I.